# Exploiting Descriptive Completeness Prior for Cross Modal Hashing with Incomplete Labels

**Haoyang Luo**
School of Computer Science and Technology
Harbin Institute of Technology, Shenzhen
Shenzhen, China
`luohaoyang.lalutte@gmail.com`

**Zheng Zhang**[*]
Harbin Institute of Technology, Shenzhen
Peng Cheng Laboratory
Shenzhen, China
`darrenzz219@gmail.com`

**Yadan Luo**
UQMM Lab, School of EECS
University of Queensland
Brisbane, Australia
`y.luo@uq.edu.au`

## Abstract

In this paper, we tackle the challenge of generating high-quality hash codes for cross-modal retrieval in the presence of incomplete labels, which creates uncertainty in distinguishing between positive and negative pairs. Vision-language models such as CLIP offer a potential solution by providing generic knowledge for missing label recovery, yet their zero-shot performance remains insufficient. To address this, we propose a novel Prompt Contrastive Recovery approach, **PCRIL**, which progressively identifies promising positive classes from unknown label sets and recursively searches for other relevant labels. Identifying unknowns is nontrivial due to the fixed and long-tailed patterns of positive label sets in training data, which hampers the discovery of new label combinations. Therefore, we consider each subset of positive labels and construct three types of negative prompts through deletion, addition, and replacement for prompt learning. The augmented supervision guides the model to measure the completeness of label sets, thus facilitating the subsequent greedy tree search for label completion. We also address extreme cases of significant unknown labels and lack of negative pairwise supervision by deriving two augmentation strategies: seeking unknown-complementary samples for mixup and random flipping for negative labels. Extensive experiments reveal the vulnerability of current methods and demonstrate the effectiveness of PCRIL, achieving an average 12% mAP improvement to the current SOTA across all datasets. Our code is available at github.com/E-Galois/PCRIL.

## 1   Introduction

Cross-modal hashing (CMH) [2, 14, 17, 11, 23, 25] addresses the highly demanding cross-modal similarity search in both web search systems and academic domains [10, 13]. By efficiently transforming multi-modal data (images, text, *etc.*) into a collection of compact binary codes, CMH maintains high-dimensional cross-modal semantic similarity while significantly minimizing computational and storage costs. Deep cross-modal hashing has achieved remarkable development by leveraging dual-stream networks or semantic encoding branches based on semantic labels [14, 17]. However,

---

[*]Corresponding author

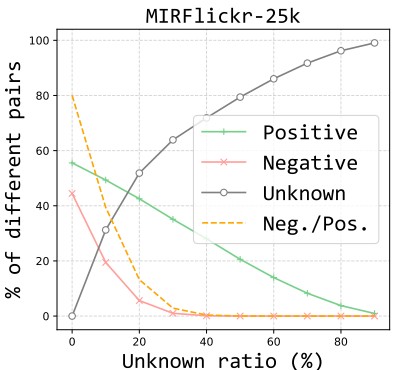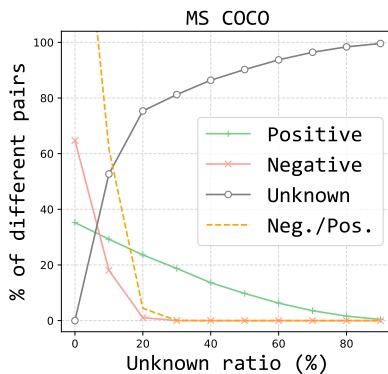

Figure 1: Incomplete labels can severely damage cross-modal similarity learning by reducing paired samples. For MIRFlickr-25k annotations (left), 35% unknown classes can completely exclude all negative pairs. For MS COCO (right), even 20% labels under-annotated can fundamentally remove the negative relationships.

due to limited labour resources, fully supervised annotation becomes impractical for large-scale datasets [27, 33]. A realistic compromise would be partial annotation, in which only a subset of classes are explicitly labeled, while others are marked as *unknown*. Although facilitating adaptation to large datasets, this labeling scheme provides significantly reduced semantic supervision. Therefore, learning with partial annotation has become a significant challenge in multi-label learning tasks.

Furthermore, CMH with partial labels inevitably encounters disrupted similarity learning due to pairwise uncertainty [27, 22]. Jointly missing classes cause incomplete labels to not only obscure class knowledge but also eliminate pairwise similarity by introducing *unknown pairs*. Among these, negative pairs constitute the majority and become particularly scarce with the increasing frequency of unknown labels. As shown in Fig. 1, negative pairs are completely removed even with 35% unknown labels in Flickr dataset. Therefore, such pairwise uncertainty can severely impair similarity-preserving hashing. However, existing CMHs desperately rely on clear positive and negative pairs to maximize supervision. It remains crucial yet unresolved how to accurately perceive potential categories and restore similarity learning in the task of CMH to prevent semantically uninformative binary codes.

To address the issues of incompleteness, some multi-label image recognition methods have investigated a feasible solution, *i.e.*, leveraging prior knowledge in large vision-language models [19, 18, 15, 24]. Pre-trained with a contrastive loss on massive image-text pairs, CLIP [28] has shown its capability to align the global representation of visual and textual modalities with loaded prior knowledge (modal correspondence, similarity structures, partial semantics, *etc.*). Although effective in many downstream tasks, they are hardly applicable for pairwise similarity recovery in CMH. Original CLIP's ability is empirically ineffective for label recovery via class-sample similarity scores. As revealed in Sec. 4.4, the original CLIP prompt yields an unsatisfactory 68% recovery precision even on the easier Flickr 30% unknown case, introducing substantial noisy classes and even degrading the final performance. Therefore, CLIP label recovery for deep CMH remains under-explored.

In this paper, we seek to overcome the aforementioned deficiencies and propose Prompt Contrastive Recovery for cross-modal hashing with Incomplete labels (PCRIL) by considering the CLIP prior knowledge of *descriptive completeness*, which is the ability to measure the completeness of a text caption for its described image. By selecting anchor class sets, we develop a simple learnable prompt to encode selected anchor class combinations into CLIP text embeddings. Multiple negative variants are constructed via editing operations on the anchor set. A prompt contrastive recovery paradigm then imposes separation gaps between these label subsets. By learning scores conditioned on their sample modalities, instance-aware class perception is enabled. Thereafter, potential labels are recovered via a tree search on the learned scores. To enhance hard incomplete samples, different instances' features and unknown labels are complementarily mixed. We also introduce an adaptive negative masking strategy to deal with negative pair scarcity at high missing rates. The main contributions of our work are as follows:

• We propose a PCRIL framework, which jointly performs semantic recovery and pairwise uncertainty elimination for efficient cross-modal hashing with incomplete labels. To the best of our knowledge, there is no prior study on CLIP-based label recovery strategies in cross-modal hashing.

• A novel recovery architecture is proposed to recover the neglected semantic labels and pairwise similarities. Particularly, a contrastive learning objective between the anchor set and its negative variants learns instance-conditioned matching scores. A tree search process then leverages the learned scores to detect potential classes. Meanwhile, a complementary semantic augmentation and an adaptive negative masking strategy jointly enhance the similarity learning in extreme cases. Thus, PCRIL can fully restore potential labels and pairwise similarity.

• Extensive experiments verify that our PCRIL can consistently outperform state-of-the-art CMH methods across a range of incompleteness levels and different benchmarks. Comprehensive analyses further validate our effective recoverability for incomplete labels.

## 2 Related Work

### 2.1 Cross-modal Hashing

Cross-modal hashing (CMH) aims to encode different high-dimensional modalities into a common space of compact binary codes where modal similarity is preserved for fast and accurate retrieval. Early machine learning methods, including those by [32, 21, 26, 3], learn common codes from encoded features. Though simple, non-deep methods are restricted by two-step training paradigm and limited in discriminative similarity learning. Jiang *et al.* [14] proposed deep cross-modal hashing (DCMH), introducing a pairwise similarity matrix into deep similarity preservation for the first time. Li *et al.* [17] constructed a label net to project labels into a common space with modal binary codes. To close modality gaps, Gu *et al.* [11] adopted cross-modal feature attentions with an adversarial learning scheme for semantic discriminability and modal consistency. Zhang *et al.* [35] proposes to preserve multi-level knowledge in CMH with a variational information bottleneck. As efficient feature learners, pretrained vision-language models have been leveraged in recent works. Tu *et al.* [30] introduced transformer-based CLIP image encoders with selective hash optimization. Liu *et al.* [23] further investigated multi-granularity cross-modal alignment based on vision-language transformers. These methods rely heavily on the guidance of complete semantic labels for similarity learning, while they fail to handle the practical incompleteness problem of label annotations in large-scale datasets.

### 2.2 Learning with Incomplete Labels

Learning with incomplete labels involves only partially known classes, which are a compromise resulting from inaccessible exhaustive supervision [7]. Existing studies regarding incomplete labels mainly focus on the image recognition task. Some works seek to recover learnability upon partial labels with modified loss functions. Bucak *et al.* [1] proposed to learn a modified ranking loss to alleviate the effects of false negatives while assuming all unknown labels to be negative. Durand *et al.* [9] proposed to learn only known labels with the prior knowledge of the label ratios. Cole *et al.* [6] proposed to solve the single positive label issue by imposing training constraints on label statistics. Another type of work investigates lost labels directly. Veit *et al.* [31] resorted to learning from human re-annotations to recover original labels. Chu *et al.* [4] proposed a variational generative model to explore data correlations with partial labels. Kim *et al.* [16] proposed to distinguish exceedingly large losses of false negatives and reverse them during training. By adopting vision-language models with rich prior knowledge, Sun *et al.* [29] proposed a prompt-tuning method with separately learned positive and negative prompt parameters. Ding *et al.* [8] further considered constructing a graph-based label structure in vision-language models to enhance image recognition. In cross-modal retrieval, the problem is however unexplored. Some methods [27] focus on the problem of entirely missing labels. Ni *et al.* [27] and Liu *et al.* [22] proposed shallow CMH methods that re-predict the labels with consistency constraints between instances and labels. However, without a fine-grained measurement of sample-label consistency, their ability for class perception is limited to only distinguish salient ones.

## 3 Proposed Approach

### 3.1 Problem Definition

We focus on image-text hashing, which is prioritized and fundamental. CMH with incomplete labels learns hash functions on a training set $\mathcal{O} = \{(\boldsymbol{f}_i, \boldsymbol{g}_i, \boldsymbol{l}_i)\}_{i=1}^N$, where $\boldsymbol{f}_i$ and $\boldsymbol{g}_i$ are the image and text

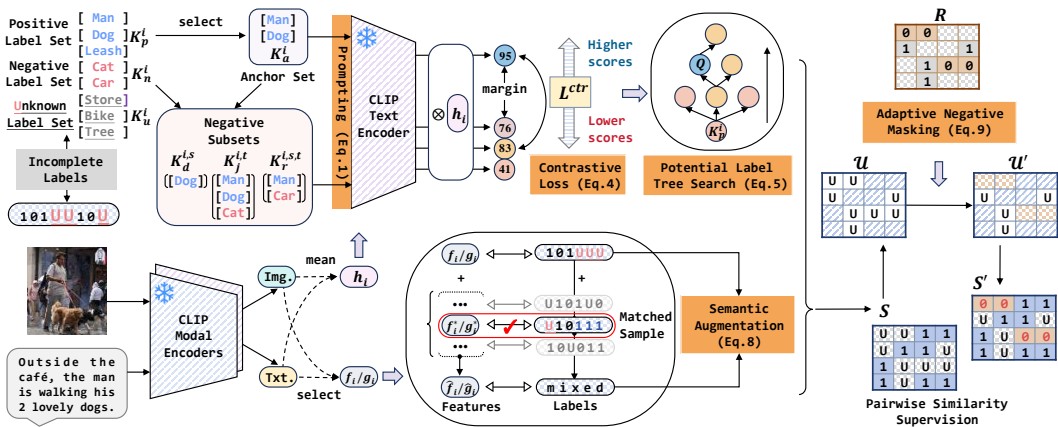

Figure 2: Our proposed PCRIL consists of two major stages: prompt contrastive recovery and augmented pairwise similarity learning. The prompt contrastive recovery stage effectively perceives incompleteness via label prompts to learn contrastive matching scores with modal samples, recovering informative semantics. The similarity augmentations further eliminate unknown labels through a complementary blending of samples and recover the scarce negative pairs using an adaptive negative masking strategy.

modalities of the $i$-th sample, respectively, and $\boldsymbol{l}_i$ is its label vector, which is the $i$-th column in the label matrix $\boldsymbol{L} \in \{1, 0, u\}^{C \times N}$. Here, $C$ is the number of classes, $N$ is the number of instances, and $u$ denotes an unknown value. Hence, a sample class could be positive (1), negative (0), or unknown ($u$). A similarity matrix $\boldsymbol{S}$ is derived from $\boldsymbol{L}$, where $s_{ij} = 1$ *iff* $\boldsymbol{f}_i$ and $\boldsymbol{g}_j$ share at least one positive label. CMH aims to learn hash functions $H^t$ and $H^v$ to encode text and image samples into binary codes $\boldsymbol{b}_i^t = \text{sign}(H^t(\boldsymbol{f}_i)) \in \{0, 1\}^d$ and $\boldsymbol{b}_i^v = \text{sign}(H^v(\boldsymbol{g}_i)) \in \{0, 1\}^d$, preserving cross-modal similarity in the Hamming space. Incomplete labels can inject uncertainty in both $\boldsymbol{L}$ and $\boldsymbol{S}$. To tackle this, we propose our PCRIL framework in Fig. 2. Herein, a novel learnable CLIP prompt for label sets is designed to recover potential positive labels in Sec. 3.2. Additionally, an unknown-complementary sample augmentation and a negative masking strategy are developed in Sec. 3.3 to deal with hard samples and negative pair scarcity, respectively.

## 3.2 Prompt Contrastive Recovery

**Prompt for Positive Anchors.** Prompt finetuning is a promising solution for lightweight model adaptation. The typical prompting method for CLIP decorates the class tokens, either with a predefined prefix [28] or a vector of learnable tokens [36, 29]. A typical handcrafted prompt template for a single unknown class can be "A photo of a CLS," where CLS is the class name. However, its fixed tokens and single-class scope hinder it from capturing inter-class complexity for label-wise recovery. In pursuit of separating labels of varied informativeness, we propose to encode label sets into CLIP embeddings.

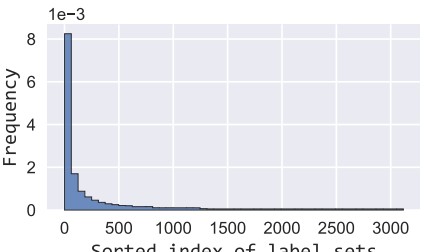

Figure 3: The sorted frequency histogram of unique positive *label sets* in MIRFlickr-25k samples at 70% known labels. This long-tail distribution induces bias for learning because many rare label combinations in the dataset are associated with limited samples.

Given a sample $(\boldsymbol{f}_i, \boldsymbol{g}_i, \boldsymbol{l}_i)$ with incomplete labels, the known classes of this instance can be extracted as the positive label subset $K_p^i = \{c \mid l_i^c = 1\}$ and the negative one $K_n^i = \{c \mid l_i^c = 0\}$. However, directly maximizing agreement between $K_p^i$ and modal representations is insufficient to capture its level of completeness. We quantified and ranked the various combinations of labels in Fig. 3. As illustrated, certain dominant combinations can overshadow less frequent cases in the current CMH scheme, obscuring their impact on model fitting. This long-tail distribution of $K_p^i$ can induce bias and disable balanced learning. Furthermore, the number of label sets available is constrained by the samples in the training dataset, which further restricts the learned correspondence. Therefore, we propose to measure semantic completeness through a contrastive learning paradigm instead, in which we consider an anchor set $K_a^i \subset K_p^i$ as

positive instance and encode it into CLIP embeddings. For instance, a typical handcrafted template can be "A photo of some `seagulls` flying above the `beach`." for $K_a^i = \{$`seagull`, `beach`$\}$ and $K_p^i$ = $\{$`seagull`, `beach`, `sky`$\}$. It is observed that class names are often surrounded by class-related prefixes and suffixes. Therefore, we further construct a learnable prompt template. The proposed prompting operation is formulated as:

$$\boldsymbol{P}(K_a^i) = (\boldsymbol{p}_{head}, \sigma(\{\boldsymbol{p}^c\}_{c \in K_a^i}), \boldsymbol{p}_{tail})$$
$$\boldsymbol{p}^c = (\boldsymbol{u}_1^c, \boldsymbol{u}_2^c, ..., \boldsymbol{u}_m^c, \text{CLS}^c, \boldsymbol{v}_1^c, \boldsymbol{v}_2^c, ..., \boldsymbol{v}_n^c), \tag{1}$$

where $\boldsymbol{p}^c$ is the specialized learnable prompt for the $c$-th class, $\boldsymbol{p}_{head} \in \mathbb{R}^{m_a \times d_f}$ and $\boldsymbol{p}_{tail} \in \mathbb{R}^{n_a \times d_f}$ are the learnable class-agnostic prefix and suffix, and $\sigma(\cdot)$ represents a random permutation operation. A specific example of the constructed prompt is given in Suppl. Sec. A.1.

**Negative Subsets and Contrastive Learning.** For the selected anchor label set, dropping any positive class object would degrade its alignment with modalities. The same is true for adding negative classes. Therefore, we aim to detect potential positive labels among the unknown ones by constructing negative subsets relative to the anchor set. Three types of negative subsets are constructed by 1) deleting a class: $K_d^{i,s} = K_a^i - \{s\}$, 2) joining a negative class in $K_n^i$: $K_j^{i,t} = K_a^i \cup \{t\}$, or 3) replacing a class by a negative one in $K_n^i$: $K_r^{i,s,t} = K_a^i - \{s\} \cup \{t\}$, where $s \in K_a^i$ and $t \in K_n^i$. Note that generating a negative set by replacement is equivalent to successively performing deletion and joining. This variant is introduced to improve model robustness.

To learn a completeness measurement, we define a simple matching score that is compatible with the CLIP prior:

$$\Phi^i(K) = E_t(P(K))^\top \boldsymbol{h}_i / \tau, \tag{2}$$

where $K$ is a label set, $E_t(\cdot)$ and $\tau$ are the CLIP textual encoder and its temperature parameter, $\boldsymbol{h}_i$ represents the average of modal CLIP features $E_v(\boldsymbol{f}_i)$ and $E_t(\boldsymbol{g}_i)$, which is analyzed in Sec. 4.4. The contrastive loss between a positive anchor set and a negative variant is formulated as

$$\mathcal{L}^i(K_a, K_*) = max(\Phi^i(K_*) - \Phi^i(K_a) + m, 0), \tag{3}$$

where $K_*$ is a negative variant of anchor set $K_a$ and $m$ is a margin parameter that separates sets of different completeness levels. The overall contrastive loss is derived as

$$\mathcal{L}^{ctr} = \sum_{i=1}^{N} \sum_{K_a^i \subset K_p^i} \left( \sum_{s \in K_a^i} \mathcal{L}^i(K_a^i, K_d^{i,s}) + \sum_{t \in K_n^i} \mathcal{L}^i(K_a^i, K_j^{i,t}) + \sum_{s,t} \mathcal{L}^i(K_a^i, K_r^{i,s,t}) \right), \tag{4}$$

Notably, the only parameter introduced for contrastive learning is the learnable prompts.

**Potential Label Tree Search (PLTS).** Through contrastive learning between label sets, knowledge in the scores is supposed to generalize as a hierarchical measurement to distinguish potential positive classes in the unknown set $K_u^i$, *i.e.*, an anchor (positive) set can be a negative set for another larger anchor set. Therefore, a simple greedy search is designed to recover potential positive labels. The label search starts with the entire positive set $K_p^i$ as a node with score $\Phi^i(K_p^i)$. At each step, we search for an unknown class that maximizes the class set score, and merge it into the positive set. Specifically, we denote the positive and unknown label sets before the $\omega$-th iteration as $K_p^i(\omega)$ and $K_u^i(\omega)$, where $K_p^i(1) = K_p^i$ and $K_u^i(1) = K_u^i$. At the $\omega$-th iteration, we try adding each $c_u \in K_u^i(\omega)$ into $K_p^i(\omega)$ and find the one that maximizes the score. This process is formulated as

$$c_u^* = \arg\max_{c_u \in K_u^i(\omega)} \Phi^i(K_p^i(\omega) \cup \{c_u\}) \tag{5}$$

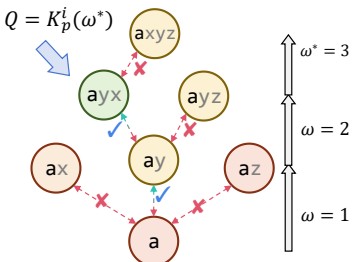

$K_p^i(1)$ : a.     $K_p^i(\omega) \dashleftarrow\dashrightarrow K_p^i(\omega) \cup \{c_u^*\}$
$K_u^i(1)$ : x, y, z.    $K_p^i(\omega) \dashleftarrow\dashrightarrow K_p^i(\omega) \cup \{c_u \neq c_u^*\}$

Figure 4: An example of the potential label tree search process. Discs represent label sets.

We then transfer the optimal $c_u^*$ into the positive label set. $K_p^i(\omega)$ and $K_u^i(\omega)$ are updated as $K_p^i(\omega + 1) = K_p^i(\omega) \cup \{c_u^*\}$ and $K_u^i(\omega + 1) = K_u^i(\omega) - \{c_u^*\}$. This process continues with the

updated sets at iteration $\omega + 1$ to further discover potential labels. The termination condition is designed as $\Phi^i(K_p^i(\omega^*) \cup \{c_u^*\}) < \Phi^i(K_p^i(\omega^*)) + \frac{m}{2}$, which ensures monotonic increase in the scores. Suppose it ends at $\omega^*$-th iteration, the search stops with a final score of $\phi = \Phi^i(Q)$, where $Q = K_p^i(\omega^*)$ is the final recovered positive set. For a specific example, Fig. 4 demonstrates a typical PLTS process. An optional choice is to further calculate a pseudo label for each of the remaining unknown classes $c_u$ by $l_i^{c_u} = H(\Phi^i(Q \cup \{c_u\}) - \phi)$, where $H(x) = \max(0, \min(1, \frac{1}{2} + \frac{x}{m}))$ is the hard sigmoid function with a linear window of $m$.

### 3.3 Augmentation Strategies for Handling Extreme Cases

Although PLTS can recover substantial positive labels, two issues can remain unresolved in highly incomplete cases. After recovery, a large portion of unknown labels can still exist. Meanwhile, negative pairs can become extremely scarce at high unknown ratios. To enhance the model for these cases, we further impose the following two augmentation strategies.

**Complementary Semantic Augmentation.** As PLTS only recovers positive labels, the unknown values cannot be fully determined, leaving uncertainty in similarity learning. Therefore, we propose to eliminate the residual uncertainty by mixing up complementary samples. However, the symmetrical mixup [34] augmentation can be ineffective because the complementary relationship is asymmetric, *i.e.*, cases that sample $\boldsymbol{f}_i$'s positive set contains an uncertain class of sample $\boldsymbol{g}_i$ but *not* vice versa. Hence, we formulate an asymmetric matrix $\boldsymbol{\Delta}^{N \times N}$ to express sample matching scores as $\delta_{i,j} = \frac{\mathcal{U}[\boldsymbol{l}_i]^\top \mathcal{P}[\boldsymbol{l}_j]}{\mathcal{U}[\boldsymbol{l}_i]^\top \mathcal{U}[\boldsymbol{l}_i]}$, where $\mathcal{U}[\cdot]$ sets unknown elements to 1 and others to 0, and $\mathcal{P}[\cdot]$ ($\mathcal{N}[\cdot]$) sets only positive (negative) entries to 1, respectively. $\delta_{i,j}$ measures the proportion of sample $i$'s unknown categories that are positive for sample $j$. This asymmetrical score evaluates the volume of semantic certainty that sample $j$ can transfer into the semantic background of sample $i$. Based on the scores, an asymmetric mix-up is introduced for a sample feature $\boldsymbol{f}_i$ and its complementary counterpart $\boldsymbol{f}_i^* = \boldsymbol{f}_j$ with label $\boldsymbol{l}_i^* = \boldsymbol{l}_j$, where $j$ is selected randomly from the top-$K$ indices on the $i$-th row of $\boldsymbol{\Delta}$. The complementary mix-up is represented as

$$\begin{cases} \hat{\boldsymbol{f}}_i = \lambda^v \boldsymbol{f}_i + (1 - \lambda^v) \boldsymbol{f}_i^* \\ \hat{\boldsymbol{l}}_i = \lambda^v \mathcal{P}[\boldsymbol{l}_i] + (1 - \lambda^v) \mathcal{P}[\boldsymbol{l}_i^*] \end{cases}, \tag{6}$$

where $\lambda^v$ ($\lambda^t$) is a learnable coefficient for the image (text) modality, determining the asymmetry in the complementary mix-up. The formula for the text modality is similar. Through this augmentation, pairwise relationship $s_{ij}$ is adjusted as deterministic values by $s_{ij} = 1 - \prod_{c=1}^C (1 - \hat{l}_i^c \hat{l}_j^c)$.

**Adaptive Negative Masking.** As illustrated in Fig. 1, negative pairs can disappear drastically at high unknown ratios. The existence of negative pairs is especially vulnerable yet significant in learning robust cross-modal relationships. Naive solutions such as AN (Assume Negative) [6] set all unknown values to 0, introducing excessive noisy pairs that can mix up with the true negative pairs and inhibit the model fitting. Therefore, we propose adaptive negative masking to restore a correct number of negative pairs. Through semantic augmentation, a real-valued similarity matrix $\boldsymbol{S}^D \in ([0, 1] \cup \{u\})^{D \times D}$ is constructed between unknown pairs with batch size $D$. By denoting $\mathcal{U} = \mathcal{U}[\boldsymbol{S}^D]$ and the ratio of $S_{ij}^D = 0$ to $S_{ij}^D > 0$ as $r = |\mathcal{N}[\boldsymbol{S}^D]|/|\mathcal{P}[\boldsymbol{S}^D]|$, the similarity matrix is adjusted adaptively as

$$\boldsymbol{S}^{D*} = \begin{cases} \boldsymbol{S}^D & r \geq t \\ (1 - \mathcal{U}) \circ \boldsymbol{S}^D + \boldsymbol{R} \circ (\mathcal{U} \circ \boldsymbol{S}^D) & r < t \end{cases}, \tag{7}$$

where $\circ$ denotes the Hadamard product, $\boldsymbol{R} \in \{0, 1\}^{D \times D}$ is a random mask that sets unknown values to 0 for a number that resets $r = t$, and $t$ is a threshold close to 0 that prevents noisy similarity structure while slightly enabling AN when there are few negatives. To learn hash codes with these augmentation strategies, the overall hash optimization is defined in Appendix Sec. A.2

## 4 Experiments

### 4.1 Experiment Settings

We evaluate our method on the MIRFlickr-25k (Flickr) [12], MS COCO (COCO) [20], and NUS-WIDE (NUS) [5] datasets. We use the frozen ViT-B-32 CLIP as backbones for all methods and set

Table 1: The MAP comprisons on Flickr, NUS, and COCO datasets with state-of-the-art CMH methods by different known ratios (30%, 50%, and 70%). We report performance rises in **red** compared to the second-best results. *: cited results with their original experiment settings. Our proposed PCRIL significantly outperforms both deep and non-deep CMH methods, verifying the ability to recover efficient similarity learning.

| Dataset | Method | 30% known labels | | | 50% known labels | | | 70% known labels | | | Mean |
|---|---|---|---|---|---|---|---|---|---|---|---|
| | | i→t | t→i | Mean | i→t | t→i | Mean | i→t | t→i | Mean | |
| Flickr | DCH [32] | 69.8 | 65.9 | 67.8 | 75.7 | 70.2 | 72.9 | 77.5 | 72.1 | 74.8 | 71.9 |
| | SDMCH [26] | 64.3 | 67.2 | 65.8 | 66.0 | 73.9 | 70.0 | 69.5 | 76.0 | 72.8 | 69.5 |
| | SCRATCH [3] | 75.8 | 68.7 | 72.2 | 82.1 | 74.6 | 78.3 | 85.0 | 77.8 | 81.4 | 77.3 |
| | WCHash [22]* | - | - | - | - | - | - | 62.5 | 62.6 | 62.6 | - |
| | DCMH [14] | 63.0 | 65.2 | 64.1 | 67.4 | 70.2 | 68.8 | 71.3 | 74.5 | 72.9 | 68.6 |
| | SSAH [17] | 58.8 | 67.6 | 63.2 | 69.2 | 73.3 | 71.3 | 75.3 | 77.4 | 76.4 | 70.3 |
| | AGAH [11] | 59.8 | 63.4 | 61.6 | 78.4 | 76.6 | 77.5 | 84.1 | 79.2 | 81.6 | 73.6 |
| | DCHMT [30] | 64.1 | 64.0 | 64.0 | 78.3 | 75.6 | 76.9 | 81.0 | 80.0 | 80.5 | 73.8 |
| | PCRIL (ours) | **78.5 (2.7)** | **75.4 (6.7)** | **77.0 (4.8)** | **85.4 (3.3)** | **79.4 (2.8)** | **82.4 (4.1)** | **87.5 (2.5)** | **82.2 (2.2)** | **84.9 (3.3)** | **81.4 (4.1)** |
| NUS | DCH [32] | 65.1 | 66.1 | 65.6 | 65.2 | 66.9 | 66.0 | 67.1 | 68.2 | 67.6 | 66.4 |
| | SDMCH [26] | 55.7 | 59.9 | 57.8 | 58.9 | 61.2 | 60.0 | 59.3 | 62.2 | 60.7 | 59.5 |
| | SCRATCH [3] | 35.5 | 64.1 | 49.8 | 28.9 | 67.4 | 48.2 | 32.6 | 68.9 | 50.7 | 49.6 |
| | DCMH [14] | 29.5 | 31.3 | 30.4 | 32.4 | 33.4 | 32.9 | 36.3 | 35.5 | 35.9 | 33.1 |
| | SSAH [17] | 35.9 | 45.3 | 40.6 | 38.4 | 57.1 | 47.8 | 46.7 | 64.0 | 55.3 | 47.9 |
| | AGAH [11] | 46.7 | 49.7 | 48.2 | 58.8 | 49.9 | 54.4 | 66.7 | 67.2 | 66.9 | 56.5 |
| | DCHMT [30] | 35.7 | 35.0 | 35.4 | 57.6 | 55.9 | 56.7 | 67.3 | 67.4 | 67.4 | 53.1 |
| | PCRIL (ours) | **67.2 (2.1)** | **70.1 (4.0)** | **68.7 (3.1)** | **68.9 (3.7)** | **70.4 (3.0)** | **69.7 (3.7)** | **70.4 (3.1)** | **72.3 (3.4)** | **71.4 (3.8)** | **69.9 (3.5)** |
| COCO | DCH [32] | 60.9 | 61.1 | 61.0 | 63.0 | 63.4 | 63.2 | 64.2 | 64.9 | 64.5 | 62.9 |
| | SDMCH [26] | 53.7 | 55.5 | 54.6 | 57.3 | 56.9 | 57.1 | 58.5 | 58.7 | 58.6 | 56.8 |
| | SCRATCH [3] | 33.5 | 59.1 | 46.3 | 34.6 | 60.9 | 47.8 | 32.6 | 63.4 | 48.0 | 47.4 |
| | DCMH [14] | 49.2 | 47.0 | 48.1 | 52.3 | 53.1 | 52.7 | 52.9 | 53.1 | 53.0 | 51.3 |
| | SSAH [17] | 32.0 | 40.4 | 36.2 | 31.1 | 50.5 | 40.8 | 36.7 | 55.6 | 46.1 | 41.0 |
| | AGAH [11] | 54.2 | 56.1 | 55.1 | 58.5 | 58.8 | 58.6 | 61.2 | 62.4 | 61.8 | 58.5 |
| | DCHMT [30] | 44.8 | 44.3 | 44.5 | 52.1 | 49.5 | 50.8 | 62.0 | 61.5 | 61.8 | 52.4 |
| | PCRIL (ours) | **62.8 (1.9)** | **63.5 (2.4)** | **63.2 (2.2)** | **64.0 (1.0)** | **64.7 (1.3)** | **64.4 (1.2)** | **67.8 (3.6)** | **68.8 (3.9)** | **68.3 (3.8)** | **65.3 (2.4)** |

Table 2: The ablation study results on Flickr, NUS, and COCO datasets. B: Baseline CMH method, IU: ignoring unobserved pair relationships, AN: assuming all unknown pairs to be negative, ANM: adaptive negative masking, PCR: prompt contrastive recovery, and CSA: complementary semantic augmentation.

| Method | Flickr | | | NUS | | | COCO | | |
|---|---|---|---|---|---|---|---|---|---|
| | 30% known | 50% known | 70% known | 30% known | 50% known | 70% known | 30% known | 50% known | 70% known |
| B w/ IU [9] | 57.5 | 73.4 | 82.8 | 62.4 | 63.3 | 67.5 | 49.6 | 50.4 | 45.9 |
| B w/ AN [6] | 68.9 | 76.6 | 81.5 | 51.1 | 53.8 | 66.2 | 45.8 | 54.3 | 59.8 |
| B w/ ANM | 75.0 | 78.1 | 83.8 | 60.6 | 60.7 | 68.1 | 59.9 | 61.4 | 65.1 |
| B w/ ANM + PCR | 76.3 | 82.1 | 84.4 | 68.0 | 69.4 | 70.9 | 62.4 | 63.7 | 67.2 |
| B w/ ANM + PCR + CSA | **77.0** | **82.4** | **84.9** | **68.7** | **69.7** | **71.4** | **63.2** | **64.4** | **68.3** |

the main hash bit as 32. Details regarding datasets, implementation, evaluation settings, and results for other bit configurations are presented in Appendix Sec. B and Sec. C.

## 4.2 CMH with Incomplete Labels

To validate the model's effectiveness on CMH with incomplete labels, we illustrate mAP comparison results in Table 1. Our model consistently achieves superior results across various known ratios on all benchmarks. We clarify that there is no prior deep CMH method specifically designed for incomplete labels. Therefore, we have to use regular CMH baselines by setting different missing ratios but keeping other settings equivalent. Compared to shallow methods, our method achieves higher results,

Table 3: Ablation results for comparison on the conventional AN setting. Results on the Flickr and COCO datasets are reported.

| Method | Flickr | | | COCO | | |
|---|---|---|---|---|---|---|
| | 30% known | 50% known | 70% known | 30% known | 50% known | 70% known |
| B w/ AN | 68.9 | 76.6 | 81.5 | 45.8 | 54.3 | 59.8 |
| B w/ AN + CSP | 68.8 | 76.2 | 82.3 | 46.4 | 54.1 | 59.7 |
| B w/ AN + PCR | 75.0 | 79.4 | 82.9 | 55.7 | 58.2 | **65.8** |
| B w/ AN + PCR + CSA | **75.3** | **80.2** | **83.5** | **58.6** | **59.6** | 65.2 |

Table 4: Prompt construction variants compared on Flickr dataset. The MAP and precisions of recovered positive labels (PRECISION) are reported. Our PCRIL can successfully marry multi-label information with CLIP prior knowledge (compared to Conventional) and yield learned prompts for instance-label matching (compared to Phrasal).

| Variant | Prompt Type | | MAP | | | | PRECISION | | | |
|---|---|---|---|---|---|---|---|---|---|---|
| | Learnable | Multi-label | 30% known | 50% known | 70% known | Mean | 30% known | 50% known | 70% known | Mean |
| Phrasal | | ✓ | 75.0 | 76.9 | 74.0 | 75.3 | 65.5 | 68.2 | 68.3 | 67.3 |
| Conventional | ✓ | | 76.3 | 81.8 | 82.8 | 80.3 | 86.0 | 89.6 | 87.0 | 87.5 |
| Ours | ✓ | ✓ | **77.0** | **82.4** | **84.9** | **81.4** | **87.4** | **89.6** | **92.0** | **89.7** |

Table 5: Prompt search variants compared on Flickr and NUS datasets. Compared to single-modal recovery, our proposed PLTS can perform instance-level matching to produce more precise results. The one-step all variant validates the effectiveness of our recursive label recovery in PLTS.

| Dataset | Variant | MAP | | | | PRECISION | | | |
|---|---|---|---|---|---|---|---|---|---|
| | | 30% known | 50% known | 70% known | Mean | 30% known | 50% known | 70% known | Mean |
| Flickr | By image | **78.2** | 79.2 | **85.3** | 80.9 | 86.2 | 86.6 | 88.1 | 87.0 |
| | By text | 74.3 | 76.6 | 84.4 | 78.4 | 70.1 | 80.2 | 77.2 | 75.8 |
| | One-step all | 64.6 | 77.4 | 82.9 | 75.0 | 21.0 | 38.4 | 54.6 | 38.0 |
| | Ours | 77.0 | **82.4** | 84.9 | **81.4** | **87.4** | **89.6** | **92.0** | **89.7** |
| NUS | By image | 51.3 | 65.4 | 68.5 | 61.7 | 78.4 | 76.0 | 74.9 | 76.4 |
| | By text | 50.8 | 65.1 | 69.3 | 61.7 | 69.4 | 69.1 | 68.9 | 69.1 |
| | One-step all | 48.4 | 64.9 | 67.5 | 60.2 | 12.1 | 23.7 | 27.1 | 21.0 |
| | Ours | **68.7** | **69.7** | **71.4** | **69.9** | **79.5** | **78.1** | **80.3** | **79.3** |

especially on Flickr, with a 4.4% improvement on average, validating the representation ability of our method. In comparison with deep CMH methods, our model enjoys greatly improved results on all three datasets, with a mean 8.96% mAP enhancement. Some deep CMH methods perform worse with incomplete labels especially on the NUS dataset, indicating their collapsed similarity learning. In comparison with DCHMT, which also adopts CLIPs as backbones, the proposed PCRIL obtains an average of 12.13% mAP improvement on all datasets. These results demonstrate that our model can consistently obtain superior results across unknown ratios and benchmarks, validating the effectiveness of our PCRIL in label recovery and similarity calibration for CMH.

### 4.3 Ablation Study

To verify the validity of each module, we conduct an ablation study of the proposed method by comparing it with 4 variants shown in Table 2 on the Flickr, NUS, and COCO datasets, and 4 module variants exclusively on the conventional AN setting. In contrast with the two conventional baselines (IU and AN), our ANM-enhanced baseline surpasses them by 6.66% and 6.09% mAP, respectively. This verifies the effectiveness of the balanced pairing strategy. By adding PCR, the model's performance is significantly improved by an average of 3.52%, validating its ability to detect incompleteness and recover labels. Although initial recovery by ANM and PCR is substantial, by adding CSA, the model further gains consistent improvements of 0.60% on average across all datasets, verifying its ability of pairwise similarity augmentation. The two modules together make a joint improvement of 4.12% on average, while the entire PCRIL rescues more than 10% of mAP from unknown labels. In ablation studies within the AN setting (Table 3), our proposed components consistently deliver stable improvements, highlighting PCRIL's ability to overcome limitations inherent in traditional training schemes. These results have comprehensively verified the effectiveness of PCRIL for CMH with incomplete labels.

### 4.4 Model Analysis

**Prompt Construction.** We analyze our proposed label prompt learning by comparing it with two variants: 1) **phrasal**: a handcrafted prompt in CLIP's original template of "A photo of $CN^{c_1}$, $CN^{c_2}$, ..., and $CN^{c_n}$." is constructed to directly perform recovery search, and 2) **conventional**: averaging single-class prompt embeddings to acquire the label prompt embeddings. The recovery results are shown in Table 4. Compared with handcrafted prompts, our learnable prompts achieve a 6.1% mAP enhancement and improve the restoration precision by 22.33%, verifying the effectiveness of

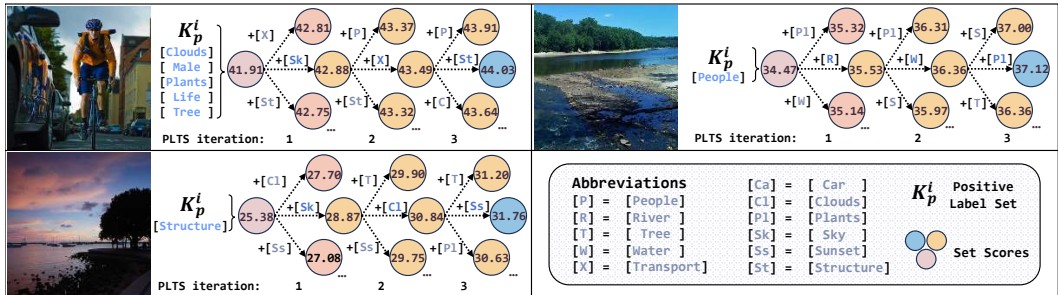

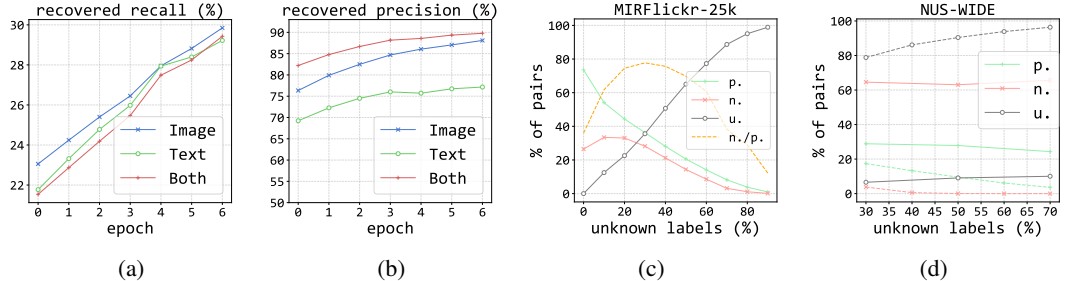

Figure 5: The recovered classes and scores of 3 case images *w.r.t.* search iterations. For brevity, we only show top-3 results at all steps. The recursive recovery of potential classes results in successive increases in the set scores.

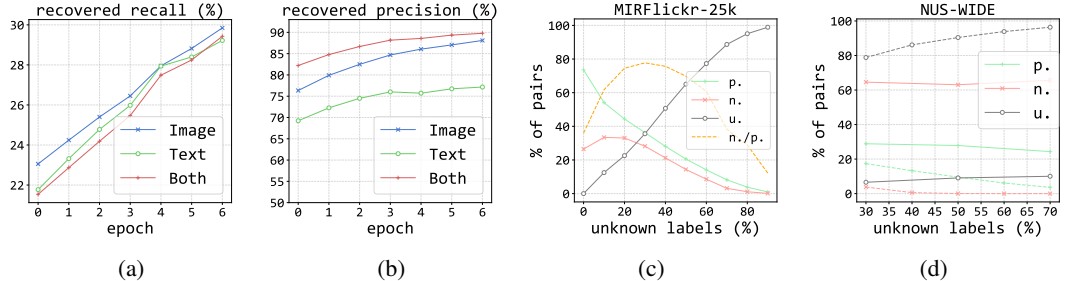

Figure 6: Recovery of labels and deterministic pairs. The left 2 subgraphs: the (a) recall and (b) precision of recovered positive classes *w.r.t* initial epochs of prompt tuning. The right 2 subgraphs: pairwise similarity recovery by (c) complementary semantic augmentation and (d) prompt contrastive recovery respectively, on the Flickr and NUS datasets. 'p.', 'n.', and 'u.' stands for positive, negative, and unknown, respectively. Dashed lines are corresponding results w/o applying our modules.

learned prompts. Compared with conventional prompts, our method achieves consistently improved performances, further validating the effectiveness of the structural prior utilized in our method.

**Prompt Search for Recovery.** We validate our prompt recovery with three variants: 1) image: label recovery search with the image modality only, 2) text: label recovery search with the text modality only, and 3) one-step all: directly recovering pseudo-labels on the initial positive label set $K_p^i(1)$ without searching, which means $Q = K_p^i(1)$. The results are shown in Table 5. Compared with image and text recoveries respectively, our recovery with both modalities displays about 3% and more than 10% improvements. This reveals our model's ability to evaluate the joint completeness in label prompts. Performing "one-step all" recovery, it hardly recalls true positives but still achieves acceptable mAPs, validating the latent hierarchical structure of the matching scores. It can be inferred that the prompt search works by peeking at scores for confident classes that aid completeness the most. Meanwhile, we plot how the number and precision of recovered positives change through the beginning epochs *w.r.t.* modalities in Fig. 6 (a-b). The precision rises with the growth of epochs while recalling more ground-truth labels, validating that the prompt gradually separates more complete and more incomplete semantic descriptors to make true positives emerge. Recovery with a single modality can introduce a drop in precision, where the text modality has especially low results of 77.16% mainly due to its nature of partial reference to labels. However, recovery with both modalities surprisingly boosts the precision to achieve a more effective 89.78%.

**Matching Scores Visualization.** To further analyze the property of descriptive completeness, we illustrate some case images' matching scores during the PLTS process in Fig. 5. In the results, the recovery of a potential positive class is accompanied by an increase in the set score, indicating that the recovered label set is more consistent with the instance. For example, the score increases from the original 25.38 to its best 31.76 by successively recalling `sky`, `clouds`, and `sunset` for the bottom-left instance. We can observe that class rankings can vary at each step, indicating that the PLTS can generate more precise measurements of multi-label complexity. Meanwhile, though not recovered as positives, the remaining scores (in orange discs) of true positives can still surpass their anchors from the last iteration, validating their effectiveness as pseudo-labels.

**Enhancement of Pairwise Similarity.** As shown in Fig. 1, CMH can suffer from severe pairwise uncertainty. By adopting our proposed prompt contrastive recovery and complementary semantic augmentation, respectively, the similarity structure is efficiently recovered in Fig. 6 (c-d). The complementary semantic augmentation greatly alleviates the pairwise scarcity. For situations at approximately 50% unknown where negative pairs are completely lost, a substantial 20% proportion of negative pairs are reconstructed, ensuring its ability to rescue similarity learning at high unknown ratios. Meanwhile, the prompt contrastive recovery significantly improves the distribution of both positive and negative pairs via its pseudo-labeling, reducing an average of 80% uncertain pairs, verifying its strong separability for completeness of labels. Generally, these two modules ensure the method can learn CMH on sufficient data pairs with clear relationships.

## 5    Conclusion

In this paper, we identified the problem of incomplete labels and the consequent collapse of similarity learning in CMH. To overcome these challenges, we introduced Prompt Contrastive Recovery for CMH with Incomplete Labels (PCRIL), a novel framework that jointly recovers semantic classes and pairwise similarity. This is the first CMH method to enable prompt learning with incomplete labels. Specifically, by constructing contrastive learning with a hierarchical label prompting method, prompt contrastive recovery learns the completeness of label descriptors and detects lost labels. Moreover, complementary semantic augmentation eliminates the sparsity of semantic pairs via a complementary feature blending strategy to restore similarity. An adaptive negative masking strategy is adopted to further balance the pairwise hashing. Extensive experiments on widely used benchmarks validated that PCRIL can significantly outperform state-of-the-art CMH methods with different partial levels.

**Limitations.** This study highlights the effectiveness of the vision-language model prior in perceiving label completeness, specifically for cross-modal retrieval. However, we note that it does have limitations. The prompt construction currently relies on the pretrained CLIP model with a limited number of text tokens, which hinders its ability for richly labeled samples. However, annotation length exceeding the CLIP capacity is extremely rare in practice. Meanwhile, the learning of PCR relies on sufficient multi-labeled samples. Single positive cases [6] may infrequently exist in real applications. We suspect that allowing unknown labels in the anchor sets could reactivate the proposed paradigm.

## Acknowledgments and Disclosure of Funding

This research is partially supported by Shenzhen Science and Technology Program (Grant No. RCYX20221008092852077), National Natural Science Foundation of China (Grant No. 62372132), and the Australian Research Council (Grant No. DE240100105, DP240101814, DP230101196).

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

# Appendix

In the following sections, we provide detailed information about our model (Sec. A) and experimental settings (Sec. B), and offer more extensive validations (Sec. C) of our proposed PCRIL.

## A  Model Details

### A.1  Constructed Prompts

We have formulated our constructed prompts in Eq. (1). Here, we describe a specific example of prompt construction. Assume we have a sample with a positive label set {cat, wall, house} and a negative one {dog, plane}. A naive textual prompt can be formulated as 'A photo of a cat, next to the wall of a house.', while our prompt is constructed by selecting an anchor subset (*e.g.*, $C_a$={wall, house}) to form $\boldsymbol{P}(C_a) = (\boldsymbol{p}_{head}, \boldsymbol{p}^{\texttt{house}}, \boldsymbol{p}^{\texttt{wall}}, \boldsymbol{p}_{tail})$ with randomly shuffled class order by $\sigma$. Then, we can directly construct two negative prompts $\boldsymbol{P}(C_n^1) = (\boldsymbol{p}_{head}, \boldsymbol{p}^{\texttt{wall}}, \boldsymbol{p}_{tail})$ and $\boldsymbol{P}(C_n^2) = (\boldsymbol{p}_{head}, \boldsymbol{p}^{\texttt{dog}}, \texttt{house}, \boldsymbol{p}^{\texttt{wall}}, \boldsymbol{p}_{tail})$ where the negative sets $C_n$={wall} and $C_n$={dog, house, wall} are formulated by adding the negative class $t = \texttt{dog}$ and removing the positive class $s = \texttt{house}$, respectively. We can also construct another negative prompt $\boldsymbol{P}(C_n^3) = (\boldsymbol{p}_{head}, \boldsymbol{p}^{\texttt{plane}}, \boldsymbol{p}^{\texttt{cat}}, \boldsymbol{p}_{tail})$ by replacing class wall with the negative one plane in $C_a$. Both positive and negative classes are uniformly selected in our experiments. We illustrate our prompt construction and contrastive learning in Fig. 7.

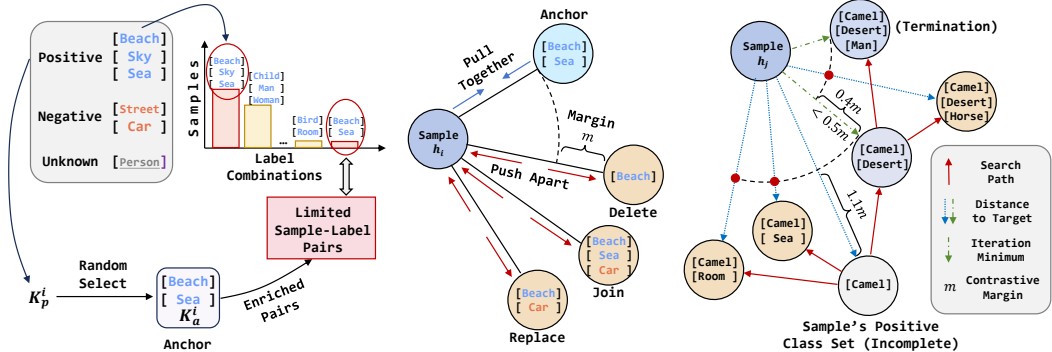

(a) The motivation of selecting positive anchor sets to enrich sample-label pairs.

(b) The relationship of the anchor and negative subsets.

(c) The potential label tree search with contrastively learned label embeddings.

Figure 7: The motivation and relationship of different components in our proposed prompt contrastive recovery.

### A.2  Optimization

To learn hash functions, the general CMH methods [14, 17] optimize the following minimum negative log-likelihood objective:

$$\min_{\theta^{v,t}, \boldsymbol{B}} -\log\ p_m(\boldsymbol{S}|\boldsymbol{\Phi}; \theta^{v,t}) + \mathcal{L}^q$$
$$\text{s.t. } \boldsymbol{B} \in \{-1, 1\}^{r \times N}, \tag{8}$$

where $\mathcal{L}^q = \|H^v(\boldsymbol{F}) - \boldsymbol{B}\|_F^2 + \|H^t(\boldsymbol{G}) - \boldsymbol{B}\|_F^2$ is the quantization terms and $H^{v,t}$ are the visual and textual hash layers. By defining the conditioned probability of the similarity as

$$p_m(s_{ij}|\phi_{ij}; \theta^{v,t}) = \begin{cases} \delta(\phi_{ij}), & s_{ij} = 1 \\ 1 - \delta(\phi_{ij}), & s_{ij} = 0 \end{cases}, \tag{9}$$

where $\phi_{ij} = \frac{1}{2}(H^v(\boldsymbol{f}_i))^\top (H^t(\boldsymbol{g}_j))$ and $\delta(\phi) = \frac{1}{1+e^{-\phi}}$, the objective function learns to align the inner-product similarity of hash codes with the ground-truth similarity values.

However, the estimated ground-truth similarity in our work, either generated by recovered pseudo labels or by complementary blending, does not apply to the above probability due to its non-binary

---
**Algorithm 1** Optimization Algorithm for PCRIL
---
**Require:** Training dataset $\mathcal{O}$.
**Ensure:** Optimal discrete hashing codes $\boldsymbol{B}$, prompt variables $\boldsymbol{\theta}^p$, modal hash-net parameters $\boldsymbol{\theta}^v, \boldsymbol{\theta}^t$.
  1: **Initialization**
     Randomly initialize learnable parameters.
     Configure the learning-rate $\mu$; the mini-batch size $D$; iterations $EP_{p.net}$ and $EP_{hash}$.
  2: **while** $iter_i < EP_{p.net}$ & not converged **do**
  3:     Sample a minibatch $(\boldsymbol{f}_i, \boldsymbol{g}_i, \boldsymbol{l}_i)_{i=1}^D$;
  4:     Sample an anchor set $K_a^i$ for each label $\boldsymbol{l}_i$ and generate its negative sets $K_d^{i,s}$, $K_j^{i,t}$, and $K_r^{i,s,t}$;
  5:     Compute loss $\mathcal{L}^{ctr}$ and update $\boldsymbol{\theta}^p$;
  6: **end while**
  7: Recover labels and pseudo labels by prompt search stated in Sec. 3.2;
  8: **while** $iter_j < EP_{hash}$ & not converged **do**
  9:     Sample a minibatch $(\boldsymbol{f}_i, \boldsymbol{g}_i, \boldsymbol{l}_i)_{i=1}^D$;
 10:     Sample and complementarily mix up instances with Eq. (6);
 11:     Compute similarity supervision and randomly transfer unknown pairs as negatives by Eq. (7);
 12:     Compute loss $\mathcal{L}^{hash}$ and update $\boldsymbol{\theta}^{v,t}$;
 13:     Update code $\boldsymbol{B}$ with $\boldsymbol{B} = \text{sign}(H^v(\boldsymbol{F}) + H^t(\boldsymbol{G}))$
 14: **end while**
---

property. Therefore, we formulate a learnable objective by leveraging the Kullback-Leibler divergence as

$$
\min_{\theta^{v,t}, \boldsymbol{B}} D_{\text{KL}}(p_d(\boldsymbol{S}) \| p_m(\boldsymbol{S}|\boldsymbol{\Phi}; \theta^{v,t})) + \mathcal{L}^q
$$
$$
\text{s.t. } p_d(s_{ij}) = \left\{ \begin{array}{ll} \hat{s}_{ij}, & s_{ij} = 1 \\ 1 - \hat{s}_{ij}, & s_{ij} = 0 \end{array} \right.
$$
$$
\boldsymbol{B} \in \{-1, 1\}^{r \times N}, \tag{10}
$$

where $D_{\text{KL}}(\cdot \| \cdot)$ is the Kullback-Leibler divergence, and $\hat{s}_{ij} \in [0, 1]$ is the estimated ground-truth similarity. Taking Eq. (9) into account, the final hashing objective is derived as

$$
\min_{\theta^{v,t}, \boldsymbol{B}} \mathcal{L}^{hash} = \Big( \sum_{i=1}^N \sum_{j=1}^N \mathbb{I}[\hat{s}_{ij} \neq u] \mathcal{L}_{ij}^{sim} \Big) + \mathcal{L}^q
$$
$$
\text{s.t. } \boldsymbol{B} \in \{-1, 1\}^{r \times N}. \tag{11}
$$

where $\mathcal{L}_{ij}^{sim} = \log(1 + e^{\phi_{ij}}) - \hat{s}_{ij}\phi_{ij}$. The detailed batch-wise learning process for PCRIL is displayed in Algorithm 1.

# B  Experimental Settings

## B.1  Datasets

We evaluate our method extensively on three widely employed benchmarks for CMH, namely, MIRFlickr-25K (Flickr)[12], NUS-WIDE (NUS)[5], and MS COCO (COCO)[20]. Flickr consists of $20,015$ image-text pairs that belong to 24 semantic classes in which $15\%$ are annotated as positive. NUS is a large-scale cross-modal dataset with about $300,000$ pairs of images and texts in 80 categories. The most frequent 21 categories are selected to form a labeled set where $10\%$ are positive. COCO is a cross-modal dataset consisting of about $120,000$ image-text instances associated with 80 semantic classes, among which confirmed positives make up only $3.6\%$ of label elements, making it a rather harder dataset as shown in Fig. 1. As we stated in Sec. 3.1, we randomly select a given proportion of binary labels and mask them as unknown during training, and unmask them for retrieval inference.

Table 6: Data split for Flickr, NUS, and COCO in our experiments.

| Dataset | Test (query) | Dataset | Train | Total |
|---------|--------------|---------|-------|-------|
| Flickr | 2,000 | 18,015 | 18,015 | 20,015 |
| NUS | 2,085 | 193,749 | 30,000 | 195,834 |
| COCO | 5,000 | 82,081 | 25,000 | 87,081 |

Table 7: The analysis for hyperparameters' impact on the model's performance on the Flickr dataset. Evaluated parameters contain the number of complementary samples ($K$) for each sample and the value of margin ($m$) used in prompt contrastive learning and PLTS process.

| | K = 10 | | m = 1.00 | |
|------|-------|--------------|-----|-------|
| m | MAP | PRECISION (%) | K | MAP |
| 0.25 | 0.826 | 86.76% | 5 | 0.823 |
| 0.50 | **0.827** | 88.82% | 10 | 0.824 |
| 1.00 | 0.824 | 89.61% | 20 | 0.822 |
| 2.00 | 0.820 | 90.78% | 50 | 0.823 |
| 3.00 | 0.816 | 91.54% | 100 | 0.825 |
| 4.00 | 0.817 | 91.29% | 200 | **0.826** |
| 5.00 | 0.814 | **91.81%** | 500 | 0.824 |

## B.2 Data Split

For the three evaluated datasets, we select larger training subsets compared to general CMH methods to better approximate the real incomplete label scenario. The statistics of our evaluated datasets are summarized in Table 6. All experiments conducted have adopted this same configuration.

## B.3 Implementation Details

We simply adopt publicly available ViT-B-32 based CLIP backbones and frozen weights for both image and text feature encoders of our PCRIL and other compared methods. Upon them, two 4-layer MLPs with the $tanh$ function as the last activation project the 512-dimensional image and text CLIP features into 32-bit binary codes, respectively. Other layers use ReLU as the activation function. The MLP layer dimensions are 512-1,024-2,048-512-32 from input to output. The MLP parameters are initialized with normal distributions and optimized with Adam optimizers for each modality. The learning rate is $10^{-4.5}$ for image and $10^{-3}$ for text.

For the label prompt encoder, we encode the prompts as text CLIP features and only tune the parameters in the prompts. These parameters are randomly initialized with a normal distribution. For prompt learning, we use the Adam optimizer with a learning rate of $10^{-4}$. During training, we finetune the prompt and MLP parameters while keeping the weights of CLIP encoders fixed to preserve prior knowledge.

We set the number of learnable tokens, specifically $n$, $m$, $n_a$, and $m_a$, as 2 for simplicity. We empirically set $t = 0.01$ as the ANP threshold and search for the best values for the margin $m$ and the top $K$ number, which are analyzed in Sec. C.2. We run all our experiments with 2 NVIDIA 2080ti GPUs.

## B.4 Evaluation Settings

We evaluate the model effectiveness *w.r.t.* a range of known ratios. From slightly incomplete to almost untrainable, the selected ratios are 70%, 50%, and 30%. To evaluate the retrieval ability of CMH models, we leverage the mean average precision (mAP) as our evaluation metric following common CMH works. When further analyzing our method in Sec. 4.4 and Sec. C, we also show the precision of recovered positive labels as evidence.

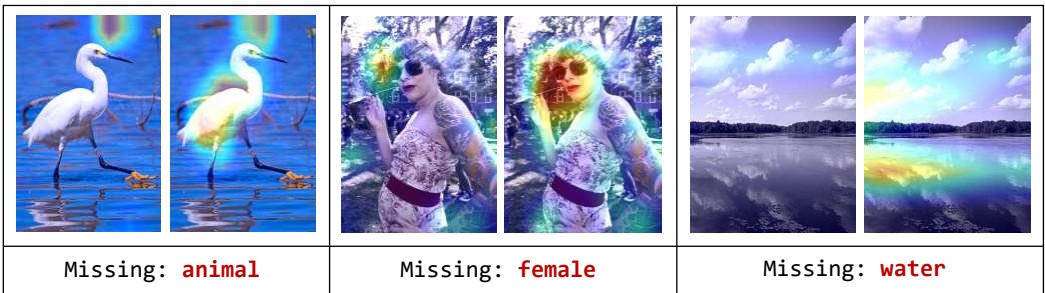

| Missing: **animal** | Missing: **female** | Missing: **water** |

Figure 8: The heatmap case visualization of the learned prompt net output. In every example: left: untrained prompts; right: trained prompts. Our learned prompts contrastively learn to attend to objects of potential classes compared to the untrained prompts.

BASELINE(FLICKR-25K)          RECOVERED(FLICKR-25K)

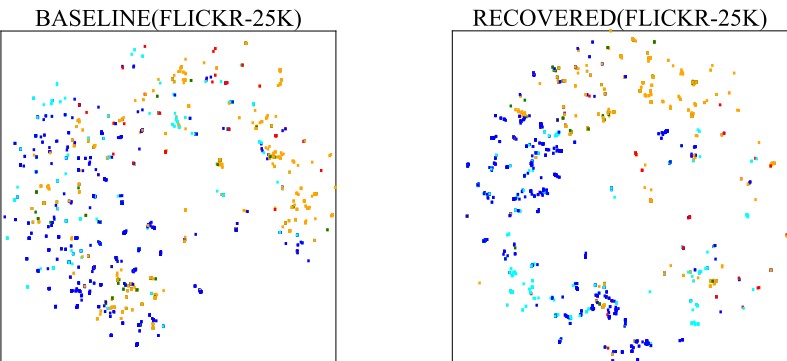

Figure 9: The t-SNE visualization for the binary codes in the baseline (AN) and our PCRIL. Colors correspond to classes. The compact clusters of semantically similar sample points verify our effective learning of discriminitive hash codes.

## C   More Experimental Analyses

### C.1   More Results for CMH with Incomplete Labels

We further validate PCRIL for CMH with incomplete labels on the IAPR TC-12 dataset. As shown in Table 8, stable improvements are achieved especially with fewer known labels, validating the effective label recovery and hash learning of the proposed method.

Table 8: Comparison results on the IAPR TC-12 dataset.

| Known ratios | 30% | | | 50% | | | 70% | | |
|---|---|---|---|---|---|---|---|---|---|
| | i→t | t→i | mean | i→t | t→i | mean | i→t | t→i | mean |
| SSAH | 31.5 | 35.2 | 33.4 | 33.1 | 45.3 | 39.2 | 42.3 | 48.3 | 45.3 |
| DCMHT | 39.1 | 39.3 | 39.2 | 48.0 | 47.8 | 47.9 | 56.7 | **55.9** | 56.3 |
| PCRIL (ours) | **45.3** | **45.6** | **45.5** | **49.2** | **48.8** | **49.0** | **56.9** | 55.4 | 56.2 |

### C.2   Hyper-parameters

We further analyze the contrastive margin parameter $m$ and the complementary blending top-$k$ in our work. The results in Table 7 demonstrate that our method is generally non-sensitive to parameter changes. Although it obtained a slightly lower recovery precision, the proposed prompt contrastive recovery can achieve its best performance of 82.7% mAP at a lower margin $m = 0.50$ by recalling more positives. Higher margins prioritize precision over recall and result in a drop in the overall mAP performance. When fixing $m$, the model achieves stable results around 82.4% mAP for different top-$K$ values.

Table 9: Comparison with SOTA CMH methods at the hash code lengths of 16, 32, and 64 on Flickr. Our PCRIL can perform consistently better across various code lengths.

| # Bits | Method | 30% known | | | 50% known | | | 70% known | | |
|---|---|---|---|---|---|---|---|---|---|---|
| | | i→t | t→i | Mean | i→t | t→i | Mean | i→t | t→i | Mean |
| 16 | AGAH | 72.0 | 69.9 | 71.0 | 79.2 | 75.7 | 77.5 | 81.1 | 77.3 | 79.2 |
| | DCHMT | 68.5 | 68.1 | 68.3 | 79.1 | 77.1 | 78.1 | 83.3 | 78.4 | 80.9 |
| | PCRIL (ours) | **77.7** | **75.6** | **76.7** | **83.3** | **77.2** | **80.3** | **86.9** | **79.6** | **83.3** |
| 32 | AGAH | 59.8 | 63.4 | 61.6 | 78.4 | 76.6 | 77.5 | 84.1 | 79.2 | 81.7 |
| | DCHMT | 64.1 | 64.0 | 64.1 | 78.3 | 75.6 | 77.0 | 81.0 | 80.0 | 80.5 |
| | PCRIL (ours) | **78.5** | **75.4** | **77.0** | **85.4** | **79.4** | **82.4** | **87.5** | **82.2** | **84.9** |
| 64 | AGAH | 61.1 | 67.2 | 64.2 | 63.5 | 66.7 | 65.1 | 84.6 | **80.9** | 82.8 |
| | DCHMT | 72.3 | 71.1 | 71.7 | 77.3 | 75.6 | 76.5 | 78.9 | 76.8 | 77.9 |
| | PCRIL (ours) | **79.5** | **76.5** | **78.0** | **85.1** | **77.7** | **81.4** | **88.5** | 80.8 | **84.7** |

## C.3   Completeness Attention Visualization

By treating each step of searching as a classification task, we list several case images in Fig. 8 with their feature heatmaps overlaid, which are generated with the ResNet-50 image backbone. From the results, we can observe that the learnable prompts successfully steer the model prior to be sensitive to potential classes. By attending more to the unlabeled object, the trained model can precisely detect incompleteness in label prompts, verifying its effectiveness in semantic detection and recovery.

## C.4   Feature Visualization

We compare the t-SNE results of learned binary features with baseline outputs in Fig. 9. In the results, our CMH learning with recovered semantics can gather hash codes of similar samples into compact clusters. Our PCRIL generates semantically more discriminative codes than the baseline method, hence achieving higher performance.

## C.5   CMH of Different Code Lengths

We compare our method with others at 16 and 64 bits in Table 9. From the results, PCRIL is validated to perform consistently well with mean mAP values across hash bits. Meanwhile, larger bits often produce higher average performance for CMH with incomplete labels, which is consistent with cases in fully labeled CMH.

