# OpenReview forum: "Exploiting Descriptive Completeness Prior for Cross Modal Hashing with Incomplete Labels"
_NeurIPS.cc/2024/Conference — NeurIPS 2024 poster_

### Official Review · Reviewer_MbLT · 2024-07-08

**Soundness:** 2
**Presentation:** 3
**Contribution:** 3
**Rating:** 7
**Confidence:** 4

**Summary:**

Cross-modal hashing (CMH) has attracted much attention due to its low computational and storage costs while maintaining high-dimensional cross-modal semantic similarity. To address the incomplete annotation challenge of CMH, this paper proposes a novel Prompt Contrastive Recovery method, PCRIL. The method includes prompt contrastive recovery (PCR) and complementary semantic augmentation (CSA) modules. Experimental evaluation verifies the effectiveness of the proposed method. Overall, this paper has done a sufficient and concrete work. However, the novelty of the proposed method is limited, and the comparison in this experiment is also limited.

**Strengths:**

1.This paper addresses the problem of incomplete annotations in cross-modal hashing, which is very common and challenging in practical applications.
2.The prompt contrastive recovery (PCR) module proposed in this paper effectively perceives incompleteness through label prompts and can well restore the information semantics, which has been well demonstrated by experiments.

**Weaknesses:**

1.The authors lack comparison with some recently proposed SOTA methods, no one within two years, so the experimental results are not convincing.
2.This paper conducted experiments on three datasets with different label known ratios, but some comparison methods provided relevant experimental results on the IAPR TC-12 dataset. The author did not provide relevant experimental results.

**Questions:**

1.There are no comparative methods within two years. Can the author perform some recent methods to show the effectiveness of the proposed method.
2.It is recommended that the authors conduct more experiments on the IAPR TC-12 dataset to more intuitively illustrate the effectiveness of the model.
3.Please elaborate on how to remove negative pairs from unknown classes.

**Limitations:**

Prompt construction currently relies on a pre-trained CLIP model with a limited number of textual labels, which hinders its ability to enrich labeled samples.

---

> ### Author Rebuttal · Authors · 2024-08-06
>
> # Authors' Responses to Reviewer `MbLT`'s Comments
>
> > Q1. The authors lack comparison with some recently proposed SOTA methods, no one within two years, so the experimental results are not convincing.
>
> We further compare our method with:
>
> - CMHML [1], a recent method investigating CMH with incomplete labels.
> - MITH [2], a recent deep CMH method which also adopt CLIP as the backbone.
>
> The results are shown in **Table 2 in the PDF file**. These results further demonstrate the superiority of the proposed approach against recent SOTAs and validate its effectiveness.
>
> [1] Ni, H., Zhang, J., Kang, P., Fang, X., Sun, W., Xie, S., & Han, N. (2023). Cross-modal hashing with missing labels. Neural Networks, 165, 60-76.
>
> [2] Liu, Y., Wu, Q., Zhang, Z., Zhang, J., & Lu, G. (2023, October). Multi-Granularity Interactive Transformer Hashing for Cross-modal Retrieval. In Proceedings of the 31st ACM International Conference on Multimedia (pp. 893-902).
>
> > Q2. This paper conducted experiments on three datasets with different label known ratios, but some comparison methods provided relevant experimental results on the IAPR TC-12 dataset. The author did not provide relevant experimental results.
>
> It's worth noting that these methods give no experimental results regarding incomplete labels. We compare our method on IAPR TC-12 with them by re-implementing their code. The results are shown in **Table 3 (PDF file)**, where our method outperforms the compared baselines significantly, especially in highly incomplete cases. This demonstrates the consistency of our method across benchmarks.
>
> > Q3. There are no comparative methods within two years. Can the author perform some recent methods to show the effectiveness of the proposed method.
> It is recommended that the authors conduct more experiments on the IAPR TC-12 dataset to more intuitively illustrate the effectiveness of the model.
>
> As stated above, we provide results of these experiments in the **PDF** file. The effectiveness of our method is validated.
>
> > Q4. Please elaborate on how to remove negative pairs from unknown classes.
>
> If we understands this correctly, the question is how to remove unknown pairs. This is achieved in our work through an **Adaptive Negative Masking (ANM)** strategy.
>
> The widely-used setting "assume negative" achieves this by turning all pairs with unknown relationship ($S_{ij}=u$) into negative ones ($S_{ij}=0$), which introduces substantial false negative sample pairs. The "only known" setting avoids pairwise learning on all unknown pairs to eliminate their influence. However, the negative relationship can disappear completely in highly unknown cases.
>
> To remove unknown pairs while eliminating the false negative side-effect, we propose to stochastically mask the unknown entries as negative to restore a balanced ratio $r=|\mathcal N(S^D)|/|\mathcal P(S^D)|$ for positive and negative pairs. Empirically, $r$ is a small positive number to alleviate introducing false negatives. In such a way, we augment the final similarity supervision $S$ with less recovered negative values but robust and higher performance.

---

> > ### Comment · Reviewer_MbLT · 2024-08-08
> > **Thanks to the authors for the response**
> >
> > Thank you for your prompt and comprehensive response to my review. Your detailed answers have effectively addressed my concerns. The additional experiments are a valuable addition, significantly strengthening the overall contribution of the paper. In addition, the proposed method is novel and well-explained. To the best of my knowledge, this is the first work that studies on CLIP-based label recovery strategies in cross-modal hashing. I am confident that this paper makes a significant contribution to the field of cross-modal hashing with incomplete labels and will be of great interest to the community. Therefore, I am raising my rating to Accept.

---

> ### Author Response · Authors · 2024-08-08
> **Thanks to reviewer MbLT**
>
> Thank you very much for your positive feedback and for raising our rating. We greatly appreciate your thorough review and the constructive suggestions regarding our experiments. We are delighted that our responses and additional experiments have effectively addressed your concerns. Your recognition of the novelty and significance of our work is highly encouraging. Thank you again for your invaluable review.

---

### Official Review · Reviewer_zzeX · 2024-07-12

**Soundness:** 3
**Presentation:** 3
**Contribution:** 3
**Rating:** 6
**Confidence:** 5

**Summary:**

This paper presents a novel cross-modal hashing method named PCRIL, which explores the indispensable but challenging problem of incomplete label recovery in multi-label learning. It conceives a CLIP-based prompting scheme and a complementary semantic propagation mechanism, enabling PCRIL to restore unknown labels and calibrate pairwise similarities. The paper exhibits a strong motivation, technical soundness, and a well-structured organization. Generally, the idea of the paper is interesting, especially in constructing a learnable label prompt to perceive the missing labels in cross-modal learning.

**Strengths:**

- This paper explores the indispensable but challenging problem of incomplete label recovery in multi-label learning. The authors identify this crucial problem for cross-modal hashing and propose an effective solution for this problem. The proposed method is technically sound.

- The paper conceives a CLIP-based prompting scheme and a complementary semantic propagation mechanism, enabling the proposed method to restore unknown labels and calibrate pairwise similarities.

- The authors design a simple learnable prompt to encode class combinations into CLIP embeddings.

- The neglected semantic labels and pairwise similarities can be removed and recovered through the proposed architecture.

- Extensive experiments have been conducted on MIRFlickr-25K, NUS-WIDE, and MS COCO.

**Weaknesses:**

- The CLIP-based prompting has been intensively studied and this paper needs to make sufficient analysis and experimental validation of its key contribution, i.e., the three types of contrastive learning, particularly in terms of their advantages over plain prompt contrastive recovery.

- Can the authors analyze the effects of the ways of the prompt construction in this task?

 This paper contributes a new problem and some new ideas to the CMH literature. The authors should further explain the necessity of the given research topic in practical situations.

The Method section should provide more explanation of these figures. Currently, these parts are separated, making the model difficult to follow, especially given the article's innovative points.

**Questions:**

Please refer to the weaknesses. The authors are advised to thoroughly resolve the above concerns to convince the reviewer.

---

> ### Author Rebuttal · Authors · 2024-08-06
>
> # Authors' Responses to Reviewer `zzeX`'s Comments
>
> Thank you for your thoughtful review. We are pleased that you found our approach novel and technically robust. Your positive feedback on our contrastive recovery method and extensive experiments is greatly appreciated. Your insights are invaluable to us.
>
> > Q1. The CLIP-based prompting has been intensively studied and this paper needs to make sufficient analysis and experimental validation of its key contribution, i.e., the three types of contrastive learning, particularly in terms of their advantages over plain prompt contrastive recovery.
>
> In section 4.3, we have provided extensive ablation of the prompt contrastive recovery. In Table 3, we have experimentally validated our advantage over plain 'phrasal' prompt, improving greatly upon the recovery precision and the final mAP.
>
> If contrastive learning is performed directly on the positive set, i.e., without selecting the anchor and constructing the three types of negative sets, we argue that the PLTS search can become ineffective because no margin is learned between similar but different label sets. However, substituting PLTS with a scan in the entire label space would consume substantial time.
>
> > Q2. Can the authors analyze the effects of the ways of the prompt construction in this task?
>
> In our experiments, we construct learnable prompt for class combinations and have analyzed it through comparisons with plain phrasal prompt and conventional CLIP-like single-class prompt.
>
> The **plain prompt** is constructed by stacking textual phrases of each class to form a prompt of the label set. Such static prompt is sensitive to the textual content it chooses and performs poor as validated in Table 3. Compared to the plain prompt, our learnable prompt can automatically adjust class-related content through contrastive learning, gaining much higher performance.
>
> The **single-class prompt** is a learnable version of the original CLIP prompt. However, it still performs not comparable to our method. The class relationship of multi-label data is intrinsically nonlinear. Therefore, by integrating different labels in a single CLIP text prompt, the model learns to capture complex relationship among classes, fostering precise contrastive learning.
>
> > Q3. This paper contributes a new problem and some new ideas to the CMH literature. The authors should further explain the necessity of the given research topic in practical situations.
>
> Our method addresses the challenge of cross-modal hashing (CMH) with incomplete labels. CMH extracts pairwise relationship from label annotations for efficient information retrieval across various data modalities, such as images and text. This capability is crucial for practical applications like web search engines, social media, and e-commerce platforms.
>
> In real-world scenarios, obtaining fully annotated datasets for CMH is often impractical. **User-generated tags** on websites, particularly on social media, are frequently incomplete and biased. For example, a user searching for "gluten-free vegan dessert recipes" may struggle to find relevant results because many recipes are simply tagged as "dessert" or "vegan," without indicating whether they are also gluten-free. Another instance of incomplete annotations occurs in **expert systems**, where labeling is costly, such as with medical images that are rarely fully annotated despite the practical demands for medical image retrieval.
>
> Our proposed method extends the traditional research topic of incomplete labels in recognition tasks to multi-modal and retrieval contexts, making CMH more robust against incomplete annotations.
>
> > Q4. The Method section should provide more explanation of these figures. Currently, these parts are separated, making the model difficult to follow, especially given the article's innovative points.
>
> Figure 2-4 illustrates the motivation of positive anchors for contrastive learning, the general framework of our method, and the Potential Label Tree Search (PLTS) process, respectively.
>
> **Figure 3** statistically illustrates the long-tailed distribution of the sample' unique class combinations. Such phenomenon implies that there are dominating combination cases that overwhelms common CMH models and hides other infrequent ones against model fitting. This motivates us to propose our method in **Figure 2**, which learns to reconstruct sample-label correspondence through anchor subsets selected from the entire positive label.
>
> As our main contribution, the **top-left part in Figure 2** defines a sequence of negative sets compared to the anchor. By embedding these label sets into learnable prompts, we perform contrastive learning to pull the anchor feature together with its sample while push the negative sets away from it. In this way, the contrastive objective in Eq.(3) would effectively separate similar label subsets with different completeness levels.
>
> The learned model with such ability allows us to propose the PLTS, whose process is demonstrated in **Figure 4**. Meanwhile, **Figure 2 top part** shows the relationship of PLTS and our contrastive learning. With the margin learned among label set embeddings, the PLTS recovers potential labels via a greedy search from the sample's positive set. It enforces the searched and joined positive classes to raise the score defined in Eq.(2), therefore recovering potential labels based on the contrastive model.
>
> We would integrate the above explanations to appropriate lines in the manuscript.

---

> > ### Author Response · Authors · 2024-08-12
> >
> > We deeply value your insightful feedback and recommendations, which have greatly enhanced the quality of our work. We welcome any additional thoughts and discussions you may have at any time.

---

### Official Review · Reviewer_Sq2X · 2024-07-14

**Soundness:** 3
**Presentation:** 3
**Contribution:** 3
**Rating:** 5
**Confidence:** 4

**Summary:**

The manuscript tackles the challenges of generating high quality hash codes for cross-modal retrieval in the presence of incomplete labels,  which creates uncertainty in distinguishing between positive and negative pairs. To address the issue, a novel Prompt Contrastive Recovery approach called PCRIL is proposed, which progressively identifies promising positive classes from unknown label sets and recursively searches for other relevant labels.

The proposed PCRIL framework jointly performs semantic recovery and pairwise uncertainty elimination for efficient cross-modal hashing with incomplete labels. In particular, they consider each subset of positive labels and construct three types of negative prompts through deletion, addition and replacement for prompt learning. Augmentation techniques are also derived for addressing extreme cases of significant unknown labels and lack of negative pairwise supervision. Experimental results show significant improvement in mAP of the proposed solution with respect to the current SOTA methods.

**Strengths:**

(1) The idea proposed in the paper seems to be novel and interesting to resolve the incompleteness for cross-modal hashing. Typically, prompt processing and contrastive learning are combined to formulate prompt constrastive recovery, which effectively detect the potential classes and enhance the similarity learning.
(2) Moreover, the augmentation techniques are also very helpful for handling extreme cases including unknown labels and negative pairs where an asymmetric mix-up is introduced and adaptive negative masking is devised .
(3) The experimental evaluation is detailed and sufficient by evaluating the proposed method on three datasets with hyper-parameter tuning and visualization.

**Weaknesses:**

(1) Some of the techniques are proposed lacking clear motivation. For instance, it is not very clear that why the mix-up augmentation is asymmetric.

(2) There are already related work on contrastive learning on prompt such as
(a) https://arxiv.org/pdf/2205.01308, "Contrastive Learning for Prompt-Based Few-Shot Language Learners". NAACL 2022.
(b) https://openaccess.thecvf.com/content/CVPR2023/papers/Zhang_CLAMP_Prompt-Based_Contrastive_Learning_for_Connecting_Language_and_Animal_Pose_CVPR_2023_paper.pdf
It is important to compare the proposed method to these existing work to see if there are performance gains. Also it seems that compared to reference (b), the proposed method is quite similar as they are also based on prompt contrastive learning for cross-modal retrieval or hashing  except that they are applied to different tasks (pose estimation vs hashing).

**Questions:**

As mentioned above, the paper needs to highlight and clearly states the motivation of some proposed techniques.

The discussion, comparison and clarification of current methods in this field need to be included.

**Limitations:**

From the impact, it seems the method is incremental and focuses on a relatively small scope.

---

> ### Author Rebuttal · Authors · 2024-08-06
>
> # Authors' Responses to Reviewer `Sq2X`'s Comments
>
> Thank you for your thorough review and insightful comments. We are delighted that you found our PCRIL approach innovative and effective in resolving incompleteness in cross-modal hashing. Your appreciation of our proposed prompt contrastive learning is very encouraging and appreciated.
>
> > Q1. Some of the techniques are proposed lacking clear motivation. For instance, it is not very clear that why the mix-up augmentation is asymmetric.
>
> The motivation of our asymmetric mix-up is to further **eliminate uncertainty** in the sample-label relationship, particularly for missing labels. The **original symmetrical mix-up augmentation is not designed for label-incomplete cases**. Due to the existence of unknown classes, cases where sample $\pmb x_i$ complements $\pmb x_j$ but $\pmb x_j$ does not complement $\pmb x_i$ can occur frequently.
>
> For instance, $\pmb l_i$ = (`sky`=1, `star`=0, `moon`=1, `person`=u) complements $\pmb  l_j$ = (`sky`=1, `star`=1, `moon`=u, `person`=0) because the `moon` tag in $\pmb l_i$ can eliminate the uncertain `moon`=u in $\pmb l_j$. However, $\pmb x_j$ does not fit in $\pmb x_i$ because filling the nonexisting `person` in $\pmb l_j$ to $\pmb l_i$ does not change the class `person`'s value.
>
> Therefore, this observation motivates us to propose our complementary mix-up design. The asymmetrical maching score gives $\delta_{ij}=0$ and $\delta_{ji}=1$, which solves such issues reasonably and effectively.
>
> > Q2. There are already related work on contrastive learning on prompt such as
> (a) https://arxiv.org/pdf/2205.01308, "Contrastive Learning for Prompt-Based Few-Shot Language Learners". NAACL 2022.
> (b) https://openaccess.thecvf.com/content/CVPR2023/papers/Zhang_CLAMP_Prompt-Based_Contrastive_Learning_for_Connecting_Language_and_Animal_Pose_CVPR_2023_paper.pdf
> It is important to compare the proposed method to these existing work to see if there are performance gains. Also it seems that compared to reference (b), the proposed method is quite similar as they are also based on prompt contrastive learning for cross-modal retrieval or hashing except that they are applied to different tasks (pose estimation vs hashing).
>
> We appreciate the two works that reviewer pointed out, however, we would like to emphasize that the studied tasks are different. More distinctive challenges can appear in the retrieval task with incomplete labels, e.g., vanished pairwise similarity and low-quality sample-label correspondence. Meanwhile, both our prompt construction and contrastive learning exhibit large uniqueness compared to the provided work.
>
> The work [1] improves language learners through clustering text examples of the same class. The method substitutes contextual demonstrations like `It is` with different prompts (e.g. `I think it is`) to produce different views of the example. The contrastive learning objective in this work is a widely-used SupCon [3] loss.
>
> - Compared to [1], our proposed method constructs **learnable** prompts for *the classes themselves*, in order to exploit the CLIP for classes' completeness knowledge.
> - Compared to their objective, our score-based contrastive margin loss, Eq.(4), involves insertion, deletion, and replacement, the three types of negative subsets, to separate class combinations of different completeness levels in terms of their CLIP scores. Such separation enables the model to discover potential positive labels in our partial-label scenario.
>
> The work [2] proposes to leverage language information to estimate animal pose keypoints. This work is more similar to ours since they also use learnable prompts for CLIP.
>
> - However, their **single-class textual prompt** each filled with the name of one pose is similar to the learnable version of the original CLIP prompts, while our multi-class prompt construction considers both learnability and authenticity for complex multi-label samples.
> - Their contrastive learning is performed between image feature maps and their pose prompts. Ground-truth point's feature and its corresponding prompt are considered a positive pair, while others are considered negative. In our work, we randomly select subsets of positive classes as anchor sets for each sample. The positive pairs are the sample-anchor pairs, while the negative ones are the sample and the negatively modified anchor. Because the stochasticity in selecting the anchor set, our learning scheme can produce diverse pair relationships. Therefore, one significant difference is that the pair relationship in our work is dynamical and can discover label completeness knowledge through the learning process of our contrastive objective.
>
> [1] Jian, Y., Gao, C., & Vosoughi, S. (2022). Contrastive learning for prompt-based few-shot language learners. arXiv preprint arXiv:2205.01308.
>
> [2] Zhang, X., Wang, W., Chen, Z., Xu, Y., Zhang, J., & Tao, D. (2023). Clamp: Prompt-based contrastive learning for connecting language and animal pose. In Proceedings of the IEEE/CVF Conference on Computer Vision and Pattern Recognition (pp. 23272-23281).
>
> [3] Khosla, P., Teterwak, P., Wang, C., Sarna, A., Tian, Y., Isola, P., ... & Krishnan, D. (2020). Supervised contrastive learning. Advances in neural information processing systems, 33, 18661-18673.
>
> > Q3. As mentioned above, the paper needs to highlight and clearly state the motivation of some proposed techniques.
> The discussion, comparison and clarification of current methods in this field need to be included.
>
> Please see the responses above.

---

> > ### Comment · Reviewer_Sq2X · 2024-08-08
> > **Response to the authors' feedback**
> >
> > Thanks for the detailed feedback. The responses and clarifications have addressed most of our comments. I have also read other reviewers' comments. I tend to maintain my original score.

---

> > > ### Author Response · Authors · 2024-08-09
> > > **Thanks to reviewer Sq2X**
> > >
> > > Thank you again for your thorough review. We are glad that our clarifications have addressed your questions. We respect your decision and are grateful for your valuable input, which have significantly strengthened the clarity and focus of our paper.

---

### Official Review · Reviewer_MwQQ · 2024-07-17

**Soundness:** 2
**Presentation:** 3
**Contribution:** 3
**Rating:** 4
**Confidence:** 3

**Summary:**

The authors propose a novel approach, Prompt Contrastive Recovery for Incomplete Labels (PCRIL), for cross-modal hashing with incomplete labels in this paper. They utilize a learnable CLIP prompt to encode selected anchor class combinations and employ a contrastive learning paradigm to construct multiple negative variants of the anchor set. Additionally, they introduce tree search methods for label recovery and develop augmentation strategies to handle extreme cases of unknown labels and negative pair scarcity. Extensive experiments on various datasets validate the effectiveness of their approach.

**Strengths:**

The paper demonstrates strong originality by combining a learnable CLIP prompt, contrastive learning paradigm, and tree search for label recovery in cross-modal hashing. It presents a well-founded and thoroughly tested solution, with comprehensive analysis and experiments that validate its effectiveness. The writing is clear and organized, making the methodology easy to understand. The contributions offer new insights and advancements that can benefit both researchers and practitioners in the field.

**Weaknesses:**

The paper could highlight its contributions by providing a detailed comparison with existing methods and including more recent studies in the related work section. A deeper theoretical analysis explaining the effectiveness of the proposed methods is needed.

**Questions:**

How does the reliance on the CLIP model's text token limit impact the overall performance of your approach? What are the specific challenges associated with needing sufficient multi-labeled samples?

**Limitations:**

The authors have identified and acknowledged several limitations of their work. And they have mentioned the limitation of the prompt construction relying on the pretrained CLIP model with a limited number of text tokens, but would it be beneficial for the authors to provide more details on how this limitation specifically affects the performance of their model.

---

> ### Author Rebuttal · Authors · 2024-08-06
>
> # Responses to Reviewer `MwQQ`'s Comments
>
> We sincerely appreciate your detailed review. We would thank your recognition of the originality and effectiveness of our approach, as well as your acknowledgment of our comprehensive analysis and clear presentation. Your feedback is highly valued and encouraging for our work.
>
> > Q1. The paper could highlight its contributions by providing a detailed comparison with existing methods and including more recent studies in the related work section.
>
> Existing studies in multi-label learning regarding incomplete labels mainly focus on single-modal recognition tasks. Compared with them, our method tackles missing labels in a **multi-modal problem**, i.e., Cross-Modal Hashing (CMH), in which **pairwise relationship** can also become sparse. Most of the related methods in classification tasks are hardly adopted directly into CMH due to distinct learning schemes. For instance, DualCoOp [1] learns class-level positive and negative prompts to transform their recognition task into a sample-class contrastive learning problem. The learning paradigm is designed only for the recognition task and only to enable learning with unknown labels. Instead, our method explicitly recovers potential classes and is an attempt to solve specific issues in CMH with incomplete labels, i.e., both the loss of sample-class correspondence and pairwise relationship.
>
> **Recent CMH studies** have several attempts [2-4] to solve the incomplete labels problem. However, they are all **non-deep** methods without fine-grained measurement of sample-label consistency. In contrast, our proposed method can not only consider minor cases with the selected anchor machenism, but also integrate precise multi-modal knowledge to recover the classes. Their ability for potential class discovery is limited to only distinguish salient ones.
>
> [1] Sun, X., Hu, P., & Saenko, K. (2022). Dualcoop: Fast adaptation to multi-label recognition with limited annotations. Advances in Neural Information Processing Systems, 35, 30569-30582.
>
> [2] Liu, X., Yu, G., Domeniconi, C., Wang, J., Xiao, G., & Guo, M. (2019). Weakly supervised cross-modal hashing. IEEE Transactions on Big Data, 8(2), 552-563.
>
> [3] Ni, H., Zhang, J., Kang, P., Fang, X., Sun, W., Xie, S., & Han, N. (2023). Cross-modal hashing with missing labels. Neural Networks, 165, 60-76.
>
> [4] Yong, K., Shu, Z., Wang, H., & Yu, Z. (2024). Two-stage zero-shot sparse hashing with missing labels for cross-modal retrieval. Pattern Recognition, 155, 110717.
>
> > Q2. A deeper theoretical analysis explaining the effectiveness of the proposed methods is needed.
>
> Consider a label encoder model $\mathcal M$ that produces optimal label encodings under loss function Eq.(4). Given a sample $\pmb  x_i$ encoded as $\pmb  h_i$ with its positive, negative, and unknown set $K_p(0)$, $K_n(0)$, and $K_u(0)$. For simplicity, we consider the $\omega$-th iteration of PLTS, the searched class $c_u \in K_u(\omega)$ is combined back to acquire $K_p(\omega+1) = K_p(\omega) \cup \{c_u\}$. With the assumption of $\mathcal M$'s generalizability, we can acquire $$\Phi^i(\mathcal M(K_p(\omega+1))) - \Phi^i(\mathcal M(K_p(\omega))) \ge m.$$ This is true for all omega if the termination condition is associated with the original margin $m$ rather than $\frac{m}{2}$ which we empirically choose to gain higher recall. We can further obtain $$\Phi^i(\mathcal M(K_p(\omega^*))) - \Phi^i(\mathcal M(K_p(0))) \ge \omega^* m.$$ This nonnegligible gap $\omega^*m$ implies that PLTS exploits the model $\mathcal M$'s ability to perceive and maximize the label completeness according to the score function $\Phi$, therefore effectively recovering potential classes.
>
> > Q3:
> How does the reliance on the CLIP model's text token limit impact the overall performance of your approach?
>
> We should clarify that we have already pointed out this limitation in manuscript Line 314-315. The following is a more detailed analysis for the impact of token limit.
>
> In **Flickr25K**, the most annotated dataset among the three evaluated dataset, the sample with the most classes contains 14 distinctive class annotations, which **does not exceed** the largest capacity of 14 classes within 77 tokens of the CLIP model. For **MS COCO**, the evaluated dataset with most classes, there are totally 12 samples exceeding the capacity, taking up **less than 0.014\%** of all samples.
>
> Furthermore, even learning on a rich-annotated dataset, substituting the CLIP backbone with, e.g., long-CLIP, may sufficiently expand the token capacity. Besides, learning with reduced labels further decreases the impact of such token limitation.
>
> In a nutshell, the negative impact is limited and can be eliminated with some simple changes to the model. How to overcome the token limitation is an *open problem* and we hope our analysis would inspire future work in these directions.
>
> > Q4. What are the specific challenges associated with needing sufficient multi-labeled samples?
>
> We should also clarify that we have already indicated this limitation in manuscript Line 316-317. The following is a more detailed analysis.
>
> Although the model can effectively recover labels even with high unknown proportions at **70%** as we illustrated in Table 1, it intrinsically relies on multi-label annotations to select non-trivial anchor sets. Without enough multi-labeled data (due to either the dataset itself or the high unknown proportion), it's difficult to utilize **the interactions between different labels**.
>
> In real-world retrieval tasks, multi-label insufficiency is relatively rare. At present, there is comparatively little research on this open topic, which can inspire future work for incomplete labels.

---

> ### Author Response · Authors · 2024-08-12
> **Discussion After Rebuttal**
>
> Thank you again for the time, thorough reviews, and constructive suggestions, which inspire us a lot for future work.
>
> Based on your comments, we provided the responses, clarifications, as well as theoretical and experimental comparisons with current research on this topic.
>
> Due to the approaching ddl of the author-reviewer discussion, we hope to further discuss with you whether your concerns have been addressed or not. If you still have any unclear parts of our work, please let us know. Thanks.

---

### Official Review · Reviewer_FiXs · 2024-07-17

**Soundness:** 3
**Presentation:** 3
**Contribution:** 3
**Rating:** 5
**Confidence:** 4

**Summary:**

In this paper, to solve the problem of unknown labels in the task of cross-modal retrieval, the authors aim to progressively identifies promising positive classes from unknown label sets and recursively searches for other relevant labels.

**Strengths:**

(1)Compared with existing related works, the proposed method has a large performance improvement.
(2)The author proposed a new strategy to solve the problem of label missing in thr cross-mdoal retriveval task.

**Weaknesses:**

(1)The meaning of the figure 3 is unclear. What is the meaning of “sorted index of label sets?” what is it relation to the positive label subset of each sample? How can we draw the conclusion that “the number of label sets is limited to the number of training samples.” from figure 3?

(2)What is the motivation and theory of the Negative Subsets and Contrastive Learning?  For the first type of negative subsets, why the anchor set is changed to negative subset by deleting a positive label and the difference between $K_a^i$ and $K_d^{i,s}$ should be minimized in the loss function of equation 4?

(3)In Potential Label Tree Search section, how to compute the score $Phi$? What is the theory of the termination condition? How to get this observation?

(4)The content is somewhat disjointed of the positive anchors and the PLTS.

(5)The ablation study setting is not reasonable. As can be seen, ANM and CSA are two augmentation strategies for handing extreme cases. The main  contribution of the paper is the Prompt Contrastive Recovery. Thus, variant version like B w/PCR, B w/PCR+CSA, B w/CSA should also be explored.

**Questions:**

Please see aboved  weeknesses.

**Limitations:**

The explanation of the main contribution part should be improved. The authors should give more motivation and related theory to verify its soundness.

---

> ### Author Rebuttal · Authors · 2024-08-06
>
> # Responses to Reviewer `FiXs`'s Comments
> Thank you for your thoughtful review and positive feedback. We are pleased that you recognize the improvements and the effectiveness of our method. Your comments are greatly appreciated and please find our point-to-point response below
>
> > Q1. The meaning of the figure 3 is unclear. What is the meaning of “sorted index of label sets?” what is its relation to the positive label subset of each sample? How can we draw the conclusion that “the number of label sets is limited to the number of training samples.” from figure 3?
>
> **(We illustrate our following points in the PDF Figure 1 for better clarity.)**
>
> **Meaning of Figure 3.** Please find our explanation to the meaning of Figure 3 in the global response. Suppose annotation $l_i = (1,1,0,u)$ indicates the presence of the first 2 classes `sky` and `sea`. The positive label set is therefore $K^i_p=(sky, sea)$. For Figure 3, the x-axis of the plot is sorted by the frequency of each set. Therefore, “sorted index of label sets” simply means label set index, sorted based on frequency.
>
> The Figure 3 illustrates the **long-tailed distribution** of unique positive label combinations, which means there are many **rare** label combinations in the dataset associated with **limited samples**. For instance, although the above $K^i_p$ is common, the subset $\{beach, sea\}$ without `sky` can appear very few times. The lack of training samples of rare label sets can cause **severe bias** in typical learning systems, which only align $x_i$ with $K^i_p$. Therefore, this motivates us to consider contrastive learning on anchor sets (subsets of $K^i_p$) to help recover unknown labels.
>
> > Q2. What is the motivation and theory of the Negative Subsets and Contrastive Learning? For the first type of negative subsets, why the anchor set is changed to negative subset by deleting a positive label and the difference between $K_d^i$ and $K_a^{i,s}$ should be minimized in the loss function of equation 4?
>
> The **motivation** behind using Negative Subsets and Contrastive Learning in our model aligns with our discussion in Q1 about handling rare label combinations. The anchor sets are selected randomly to help **enrich** the label-sample pairs. However, due to reduced positive classes, they do not perfectly align with the sample. Nonetheless, we can solve this dilemma by considering the missing of labels as relative *edit distances* to the full labels. For instance, when a positive tag is missing in the anchor, the *edit distances* to both the full and the anchor sets increase and the embedding (or specifically, learnable prompt) should be adjusted to reflect this change. This adjustment ensures that the model can better differentiate between similar but distinct items, improving its overall precision.
>
> To clarify a potential misunderstanding in the comment: our goal is **not** to minimize the difference between $K_d^i$ (the negative subset) and $K_a^{i,s}$ (the anchor set with positive labels). Instead, our objective **maximizes** this difference. In our loss function Eq.(4), we employ a contrastive loss strategy that seeks to **enlarge the discrepancy** between embeddings of the original anchor set and those of the negative subset. This discrepancy forces the model to create more distinct embeddings for negative subsets, thereby pushing the embedding of the anchor set with positive tags to be more accurate and distinct.
>
> > Q3. In Potential Label Tree Search section, how to compute the score? What is the theory of the termination condition? How to get this observation?
>
> **PLTS Scores.** The score $\Phi$ during PLTS is defined the same as Eq.(2) and is the same score used in Eq.(4). To compute the score $\Phi^i(K)$ for a given label set $K$ and sample $\pmb  x_i$, we first construct a prompt $P(K)$ according to Eq.(1), then compute its CLIP score with the instance by Eq.(2).
>
> **Termination Condition.** The termination condition $\Phi^i(K^i_p(\omega^*) \cup \{c^*_u\}) < \Phi^i(K^i_p(\omega^*)) + \frac{m}{2}$ is associated with $m$, which is the margin we used in the contrastive loss Eq.(3) to separate different levels of sample-label similarity scores, i.e., $\Phi$.
>
> In our global response, we explain that by optimizing Eq.(3), our method separates different label sets by a gap proportional to the edit distance. The termination condition yields completed labels with largely increased CLIP scores. In our method, we empirically set the margin *during PLTS* as $\frac{m}{2}$ to attach more importance to recall over precision.
>
> > Q4. The content is somewhat disjointed of the positive anchors and the PLTS.
>
> As also responded to comment 1-1, Figure 3 demonstrates the long-tailed distribution of samples' unique label combinations, which includes many rare cases and some dominating cases. The positive anchors are chosen randomly at each training step to enable coverage of a broader class combinations. For instance, the combination of (`sky`, `beach`, `sea`) is a common one while only (`beach`, `sea`) is quite rare. Our method selects random anchor sets that can effectively involve uncommon cases. As the anchor sets are randomly selected at each training step, this covers most class subsets during training. Due to this randomness, **an anchor (positive) set can be a negative set for another larger anchor set**. Therefore, each positive anchor found in the PLTS iteration $j$ is regarded as a negative set in the $(j+1)$-th iteration.
>
> > Q5. The ablation study setting is not reasonable. As can be seen, ANM and CSA are two augmentation strategies for handling extreme cases. The main contribution of the paper is the Prompt Contrastive Recovery. Thus, variant version like B w/PCR, B w/PCR+CSA, B w/CSA should also be explored.
>
> We provide ablation results with B w/PCR, B w/PCR+CSA, and B w/CSA in Table 1 in the PDF file. Stable improvements by our proposed components have been made to the default AN setting.

---

> ### Author Response · Authors · 2024-08-12
>
> Thank you for the valuable comments and suggestions. We are encouraged that you appreciated our contributions including the novel strategy and large performance improvements.
>
> Since there is limited time left for discussion, if you have any other questions, we would like to provide further clarifications and discussions about this work. Any discussion is welcome. Thanks.

---

> ### Comment · Reviewer_FiXs · 2024-08-13
>
> Thanks for your response. I incline to keep my initial score.

---

### Author Rebuttal · Authors · 2024-08-06

# Global Reply to all reviewers

We would like to extend our gratitude to all the reviewers for their insightful comments and unanimous acknowledgement of our paper in the following aspects:
1. The task addressed by this work is both interesting and significant for real-world cross-modal hashing applications (zzeX, MbLT).
2. The proposed PCRIL method has clear motivation and strong novelty by introducing prompt-based contrastive learning to perceive incomplete classes for the task (MwQQ, Sq2X, zzeX).
3. The paper is clearly written, well-structured, and easy to understand (MwQQ). Our prompt contrastive recovery is effective in addressing the challenge of incomplete labels in cross-modal hashing (Sq2X, MbLT).
4. It shows substantial performance improvements over existing methods (FiXs) and our contributions has been validated with extensive and convincing experiments(MwQQ, Sq2X, zzeX).
5. The contributions provide new insights and advancements that benefit future research in the field (MwQQ).

It is worth noting that the **PDF file** contains explanations of the **motivations and interrelationships of each component** in our method (Figure 1), as well as **the latest experimental results** including extended ablation results, comparison with recent SOTA methods, and results on a new dataset IAPR TC-12 (from Table 2 to 4, respectively).

In addition, we hereby provide highlights to some common queries regarding our current work.

1. **Motivation from Figure 3**. For cross-modal retrieval, each multi-modal instance is associated with a label annotation within a predefined class set. Figure 3 is a statistical analysis of **unique** positive label combination patterns in all samples' annotations. This illustrates the long-tailed distribution of class combinations. For rare label cases, sample-label pairs hardly exist in the data. Our proposed contrastive anchors can enrich this relationship and help label recovery in all cases.
2. **Theory for PLTS effectiveness**. If the objective Eq.(3) is fully minimized, the scores can carry label completeness information. By separating class sets that have edit distance $D$ by a large gap $G=Dm$, The trained model guarantees to increase the CLIP score from the original incomplete label by $G$ through the potential label tree search process, which effectively perceives potential classes from the unknown set.

We have provided detailed responses to each reviewer's feedback including these questions. Please find our point-to-point responses in the individual replies.

**References** for the PDF file.

[1] Ni, H., Zhang, J., Kang, P., Fang, X., Sun, W., Xie, S., & Han, N. (2023). Cross-modal hashing with missing labels. Neural Networks, 165, 60-76.

[2] Liu, Y., Wu, Q., Zhang, Z., Zhang, J., & Lu, G. (2023). Multi-Granularity Interactive Transformer Hashing for Cross-modal Retrieval. In Proceedings of the 31st ACM International Conference on Multimedia (pp. 893-902).

---

### Comment · Area_Chair_2JFa · 2024-08-14

Dear all,

Thanks for your time and efforts in reviewing this paper. This is the right and emerging time to discuss this paper with the authors.

The authors provided their rebuttal, and some reviewers have posted their partial discussion about this paper.

Any discussion is welcome and you may consider reading each others' reviews, posting a question, and reaching a consensus.

Best,
Your AC

---

### Decision · Program_Chairs · 2024-09-25

**Decision:**

Accept (poster)

**Comment:**

This paper proposes a prompt contrastive recovery approach that progressively identifies promising positive classes from unknown label sets and recursively searches for other relevant labels. Several experiments on benchmark datasets show that they significantly perform favorably against several methods. Four of five reviewers rate the paper as weak/boderline accept, while one reviewer still has concerns about the limitation of the pre-trained CLIP model. After discussion, the final decision is accept, and the authors should further explore its limitations and solutions.